# Knowledge-Augmented Reasoning Distillation for Small Language Models in Knowledge-Intensive Tasks

**Minki Kang**[1,2,*], **Seanie Lee**[2], **Jinheon Baek**[2], **Kenji Kawaguchi**[3], **Sung Ju Hwang**[2,4]

[1]KRAFTON, [2]KAIST, [3]National University of Singapore, [4]DeepAuto.ai

{zzxc1133, lsnfamily02, jinheon.baek}@kaist.ac.kr,
kenji@comp.nus.edu.sg, sjhwang82@kaist.ac.kr

## Abstract

Large Language Models (LLMs) have shown promising performance in knowledge-intensive reasoning tasks that require a compound understanding of knowledge. However, deployment of the LLMs in real-world applications can be challenging due to their high computational requirements and concerns on data privacy. Previous studies have focused on building task-specific small Language Models (LMs) by fine-tuning them with labeled data or distilling LLMs. However, these approaches are ill-suited for knowledge-intensive reasoning tasks due to the limited capacity of small LMs in memorizing the knowledge required. Motivated by our theoretical analysis on memorization, we propose **K**nowledge-**A**ugmented **R**easoning **D**istillation (**KARD**), a novel method that fine-tunes small LMs to generate rationales obtained from LLMs with augmented knowledge retrieved from an external knowledge base. Moreover, we further propose a neural reranker to obtain documents relevant to rationale generation. We empirically show that KARD significantly improves the performance of small T5 and GPT models on the challenging knowledge-intensive reasoning datasets, namely MedQA-USMLE, StrategyQA, and OpenbookQA. Notably, our method makes the 250M T5 models achieve superior performance against the fine-tuned 3B models, having 12 times larger parameters, on both MedQA-USMLE and StrategyQA benchmarks.

## 1 Introduction

Large Language Models (LLMs) [5, 8] have excelled at various tasks across diverse domains with in-context learning. Recently, scaling up the number of parameters of LLMs has been shown to significantly improve their knowledge encoding and reasoning capability [54, 24]. Moreover, such LLMs have achieved remarkable performance on knowledge-intensive tasks in professional domains which are highly challenging, since they require a considerable depth of domain knowledge and reasoning [34, 48]. For example, in Figure 1 top, answering a medical question requires both domain knowledge and reasoning ability. The LLM should understand that the patient likely has ALS based on the symptoms and recognize SOD1 is the main cause of motor neuron diseases. Furthermore, it needs to reason over the knowledge that a mutation in SOD1 is highly associated with the symptoms.

Despite its effectiveness, deploying LLMs can still be challenging, especially in real-world applications. Firstly, utilizing LLMs to make predictions is computationally expensive. It requires 326GB GPU memory to load the GPT3-175B model [11]. Moreover, deployment of the LLM potentially poses a risk of privacy leakage since most of the production-grade LLMs [5, 44, 8, 43] operate in a black-box manner. That is, users cannot access the parameters of LLMs but only their output via some Application Programming Interfaces. Consequently, the need for *white-box Small Language Models* tailored to address problems requiring domain-specific knowledge will continue to gain

---

*Work done at AITRICS. †Code is available at https://github.com/Nardien/KARD.

**Question:** A 57-year-old man presents with a 2-month history of right upper and lower extremity weakness. (...) His right upper extremity shows forearm atrophy and depressed reflexes while his right lower extremity is hypertonic with a positive Babinski sign. Which of the following is most likely associated with the cause of this patient's symptoms? A) HLA-B8 haplotype B) HLA-DR2 haplotype **C) Mutation in SOD1** D) Viral infection

**Rationale:** The symptoms suggest motor neuron disease and among the options, mutation in SOD1 is the most common cause of inherited motor neuron diseases. A viral infection can cause peripheral neuropathy, but the symptoms described suggest a central nervous system disorder.

**Relevant Knowledge** (retrieved with Rationale)
[Superoxide dismutase] Mutations in the first SOD enzyme (SOD1) can cause familial amyotrophic lateral sclerosis (ALS, a form of motor neuron disease). (...)

**Irrelevant Knowledge** (retrieved with Question)
[HLA A1-B8 haplotype] A1-B8 serotype was associated with a number of diseases as "HL-A"' antigens were first being described. (...)

Figure 1: **Concept.** An example of a knowledge-intensive reasoning task (medical QA [23]) on the top. On the bottom, we provide the conceptual illustration of our KARD, compared to existing reasoning distillation. On the right, we provide examples of passages retrieved with rationale and question from the external KB.

prominence. To tackle the above challenges of deploying models, previous works [33, 17, 38, 12, 18] have proposed to transfer the reasoning ability of large models to small models through *reasoning distillation* (See Figure 1 left). In particular, they leverage the LLM to generate high-quality rationales and fine-tune a small LM to generate the rationale obtained from the LLM. This reasoning distillation improves the performance of small LMs on tasks that require complex reasoning ability (e.g., arithmetic and symbolic reasoning [10, 55]). Based on this observation, we pose a research question: *"Is it possible to transfer both the domain knowledge and reasoning ability of LLMs through reasoning distillation, for tasks requiring specific knowledge to reason for answering a question?"*

Existing reasoning distillation is suboptimal to solve such knowledge-intensive reasoning tasks since small, distilled LMs are limited in their capacity to memorize the knowledge that is necessary to solve the tasks due to the small number of parameters. This motivates us to develop a method that distills the reasoning ability of LLMs into smaller LMs while injecting the specific task-relevant knowledge. Specifically, we augment a small LM with the knowledge retrieved from an external Knowledge Base (KB) as a non-parametric memory, and we theoretically show that the non-parametric memory can reduce the number of bits to memorize training data for performing well.

Based on this intuition and the theoretical analysis, we propose **Knowledge-Augmented Reasoning Distillation (KARD)** which enables to transfer the reasoning ability of an LLM to a small LM while injecting the knowledge, for knowledge-intensive reasoning tasks. Specifically, we utilize a retriever [46] to obtain passages containing relevant knowledge for generating a rationale from an external knowledge base (e.g., Wikipedia). We then fine-tune the small LM to generate the rationale, obtained from the LLM, based on the question and the retrieved document, and predict the answer.

During training, using a rationale as a query helps retrieve pertinent knowledge for generating rationales. However, during inference, relying on the question as a query may result in poor retrieval. As shown in Figure 1, the passage retrieved with the question is not relevant to generating the rationale. To mitigate the issue, we introduce a *neural reranker* to prioritize passages useful for rationale generation, ensuring a retrieval of relevant documents even with the question as the query.

To verify the efficacy of KARD, we empirically show that it significantly improves the performance of small LMs (OPT [20, 59] and T5 [45, 53]) on medical Question Answering (QA) (MedQA-USMLE [23]), multi-step factual QA (StrategyQA [14]), and commonsense reasoining (Open-bookQA [39]) datasets compared to few-shot in-context learning, fine-tuning, and reasoning distillation without knowledge augmentation. Also, our extensive analyses demonstrate that our KARD is efficient in terms of both the training data and the model size. Specifically, KARD with 250M models achieves higher accuracy than the fine-tuned 3B models, and KARD outperforms the fine-tuning only with a quarter of the full training data in 780M models.

Our findings and contributions are as follows:

- We demonstrate that fine-tuning small LMs to generate rationales from large LMs is insufficient for knowledge-intensive reasoning tasks and a non-parametric external knowledge base plays a crucial role in complementing the lack of knowledge in small LMs.
- Moreover, we address the limitations of the existing retriever method by introducing a reranker, in order to obtain pertinent passages for generating rationales in knowledge-intensive reasoning tasks.

- In widely-used medical, multi-step factual, and commonsense QA benchmark datasets, we empirically show that the proposed KARD significantly improves the performance of small LMs.

## 2 Related Works

**Large Language Models** Large Language Models (LLMs) have shown impressive capabilities across various tasks. One of their notable strengths is their ability to memorize knowledge and leverage that knowledge to solve knowledge-intensive reasoning tasks. For example, LLMs like GPT-3.5 [44], Med-PaLM [48], ChatGPT [29], and GPT-4 [43] have shown the promising performance on the challenging medical question answering task, the United States Medical Licensing Examination (USMLE) [23], even surpassing the passing score by a large margin [41]. However, deploying LLMs in offline and privacy-sensitive environments is still challenging since most of these models are in black-box (accessible via APIs), and computationally expensive. Thus, we need alternative solutions that can leverage the capabilities of LLMs for knowledge-intensive reasoning tasks.

**Reasoning Distillation from LLMs** Recent works [33, 17, 38, 12, 18] have attempted to distill the reasoning ability of LLMs into small LMs, where the reasoning ability is an *emergent property* which enables LLMs to perform better in reasoning tasks through Chain-of-Thought (CoT) prompting (e.g., *Let's think step-by-step*) [28, 55]. Unlike arithmetic or symbolic reasoning tasks, however, previous works [33, 17] have shown that reasoning distillation is less effective for knowledge-intensive reasoning tasks [14] where factual knowledge is important to generate accurate rationale. Therefore, we augment small LMs with documents retrieved from the external knowledge base so that the models can leverage knowledge to generate better rationales that lead to correct answers.

**Knowledge-Augmented LMs** Knowledge-augmented LMs have utilized an external Knowledge Base (KB) to supplement their intrinsic knowledge [16, 32, 3, 22, 58]. One common approach to incorporate external knowledge is by retrieving relevant passages from a KB, such as Wikipedia, based on the input query [7]. Retrieving the correct evidence is crucial to generate accurate answers and factually grounded rationales. However, previous works usually have not explored the use of knowledge-augmented LMs for tasks that require complex reasoning over knowledge. Recently, BehnamGhader et al. [1] examined the reasoning ability of existing retrieval-augmented LMs and found that the existing retriever [25] is insufficient for retrieving relevant passages to solve the knowledge-intensive reasoning tasks. To address this limitation, we propose a re-ranker for rationale generation that prioritizes passages relevant to the rationale generated by LLMs given the query. This approach can be seen as a form of knowledge distillation for the retriever, as we use the rationale to guide the reranker to retrieve more relevant passages for reasoning, instead of using plain queries.

## 3 Motivation: Effect of Knowledge-Augmentation on Memorization

Large language models are known to memorize its training data [6, 49] and the memorization capacity is proven to increase as the size of the model increases [27, 57]. The previous work [4] showed that the memorization of training data is indeed necessary to perform well in a language problem. These results suggest that the reasoning distillation with a small language model (without knowledge augmentation) will degrade the performance because of (1) the incapability of memorizing training data and (2) the necessity of the memorization to perform well. In this section, we demonstrate that using an external Knowledge Base (KB) as a non-parametric memory with a retriever reduces the amount of the memorization needed to perform well and thus allows us to use small models.

### 3.1 Background without Knowledge-Augmentation

We adopt the exact same problem setting used in Brown et al. [4]. A task distribution $P \sim q$ is drawn from meta-distribution $q$. Given a $P$, the training dataset $X = ((Z_i, Y_i))_{i=1}^n$ and the test sample $(Z, Y)$ are drawn as $X \sim P^{\otimes n}$ and $(Z, Y) \sim P$. Here, $Z$ is the input (i.e., the sequence of symbols) and $Y$ is the label (i.e., the next symbol to be predicted). The overall error of a learning algorithm $\mathcal{A}$ on the meta-distribution $q$ with sample size $n$ is defined by

$$\mathrm{err}_{q,n}(\mathcal{A}) = \Pr_{\substack{P \sim q, X \sim P^{\otimes n}, \\ (Z,Y) \sim P}} (M(Z) \neq Y \text{ where } M = \mathcal{A}(X)).$$

Given $q$ and $n$, there exists an optimal learner $\mathcal{A}_{\mathrm{OPT}}$ that minimizes this overall error, which will be used as our reference. We adopt the abstracted language problem, i.e., the next-symbol prediction problem with $N$ reference strings $\{c_j\}_{j=1}^N$ where $c_j \sim \mathrm{Uniform}\left(\{0,1\}^d\right)$, considered in the main

text of Brown et al. [4] with no symbol corruption (see [4] or **Appendix A.1** for the details). Under this setting, Brown et al. [4] proved that any algorithm $\mathcal{A}$ needs to memorize the $nd$ bits of training data to achieve $\epsilon$-suboptimality where $I$ denotes the mutual information:

**Theorem 1** (Brown et al. [4])**.** *Let $N = n$. Then, any learning algorithm $\mathcal{A}$ that satisfies $\mathrm{err}_{q,n}(\mathcal{A}) \leq \mathrm{err}_{q,n}(\mathcal{A}_{\mathrm{OPT}}) + \epsilon$ for $\epsilon = o(1)$ also satisfies $I(X; \mathcal{A}(X)|P) = \Omega(nd)$.*

### 3.2 Memorization with Knowledge-Augmentation

In Theorem 1, $d$ corresponds to the size of KB. Thus, it shows that if the size of KB is small, then a small model can just memorize all KB by memorizing $\Omega(nd)$ information to perform well. However, if the size of KB is large, then a small model cannot memorize $\Omega(nd)$ information and hence the performance is expected to drop significantly when replacing a large model with a small model. In this subsection, we show that knowledge-augmentation reduces the memorization requirement of $\Omega(nd)$ bits to that of $O(n \log_2(N + R))$ bits, allowing the use of small models.

We consider an inference algorithm $\varphi$ that uses a KB with a non-parametric retriever as follows:

$$\mathrm{err}_{q,n}^{\varphi}(\mathcal{A}) = \Pr_{\substack{P \sim q, X \sim P^{\otimes n}, \\ (Z,Y) \sim P}} (\varphi(Z, M, S) \neq Y \text{ where } M = \mathcal{A}(X)).$$

An inference algorithm $\varphi$ has no learnable parameters and makes prediction based on both the result of learning algorithm $M = \mathcal{A}(X)$ and a KB denoted by $S$, which is defined as follows. Given a task instance $P \sim q$, we choose a KB such that $|S| = N + R$ and $\{c_j\}_{j=1}^N \subseteq S$ where $R$ is the number of extra references that are irrelevant to this task $P$; i.e., $R = 0$ in the best scenario.

Theorem 2 shows that the knowledge-augmentation reduces the amount of memorization to achieve $\epsilon$-suboptimality, from the $nd$ to $\min(N, n)m$ bits, under the same problem setting as Theorem 1:

**Theorem 2.** *There exists a pair of inference and learning algorithms $(\varphi, \mathcal{A})$ such that for any $\epsilon > 0$, $\mathrm{err}_{q,n}^{\varphi}(\mathcal{A}) \leq \mathrm{err}_{q,n}(\mathcal{A}_{\mathrm{OPT}}) + \epsilon$ and $I(X; \mathcal{A}(X)|P) = O(\min(N, n)m)$ where $m = \log_2((1 - (\frac{N-1}{N})^n)\frac{(N+R)^2 - (N+R)}{2\epsilon}) \leq \log_2(\frac{(N+R)^2}{2\epsilon})$.*

With $n = N$ and $\epsilon = o(1)$, we have $I(X; \mathcal{A}(X)|P) = O(\min(N, n)m) = O(n \log_2(N + R))$ (see **Appendix A.2** for proof). Thus, it shows that knowledge-augmentation allows the reduction from the $nd$ bits to $n \log_2(N + R)$ bits for the amount of memorization needed to perform well.

## 4 Knowledge-Augmented Reasoning Distillation

We propose Knowledge-Augmented Reasoning Distillation (**KARD**), which consists of two learning processes: (1) reasoning distillation where we leverage Large Language Models (LLMs) to generate a rationale with black-box APIs and then fine-tune small models to generate both rationale and answer given a question and knowledge, in which the knowledge is retrieved from Knowledge Base (KB) with the rationale as a query; (2) reranker training to retrieve relevant passages for the question as a query at the inference time, for generating effective rationales. Our approach is illustrated in Figure 2.

### 4.1 Teach Small Models to Generate Rationales with External Knowledge

**Rationale Generation with LLMs** In our problem setup, we assume that training dataset $((\boldsymbol{x}_i, \boldsymbol{y}_i))_{i=1}^n$ for the target task is given, where $\boldsymbol{x}_i$ is input sequence (question in QA) and $\boldsymbol{y}_i$ is label (answer in QA). Additionally, there are LLMs accessible through black-box APIs [5, 8, 44, 43, 42]. In other words, the parameters and the architecture of the LLM are unknown and we can only access text sequences generated by the LLM. Since the ability to generate high-quality rationale is known as the emergent ability of LLMs [55, 28], we want to transfer such ability to a small language model with reasoning distillation. Firstly, we leverage the chain-of-thought prompting [28] to elicit the proper $l$ rationales for each training data point with LLMs: $\boldsymbol{r}_{ij} = \mathtt{LLM}(\boldsymbol{p}, \boldsymbol{x}_i, \boldsymbol{y}_i)$ for all $i \in [n] := \{1, \ldots, n\}$ and $j \in [l]$, where $\boldsymbol{r}$ is the generated rationale and $\boldsymbol{p}$ is the chain-of-thought prompt [55, 28, 48].

**Fine-tuning Small Models on Rationales** Then we fine-tune a small language model $p_\theta$ with trainable parameters $\theta$ to generate both rationale $\boldsymbol{r}_{ij}$ obtained from the LLM and answer $\boldsymbol{y}_i$, given the question $\boldsymbol{x}_i$. In other words, we minimize the negative log-likelihood of the sequence of rationale $\boldsymbol{r}_{ij}$ and the answer $y_i$ where the rationale must be generated first prior to the answer generation:

$$\mathcal{L}_{\mathtt{distill}}(\theta) = -\frac{1}{n \cdot l} \sum_{i=1}^n \sum_{j=1}^l \log p_\theta(\boldsymbol{r}_{ij}, \boldsymbol{y}_i | \boldsymbol{x}_i). \tag{1}$$

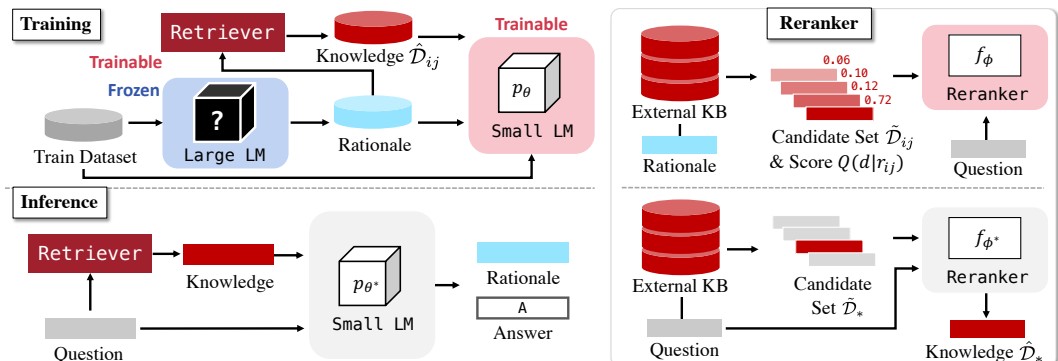

Figure 2: **Overview of KARD.** (Left, § 4.1) Illustration of training (top) and inference (bottom) of knowledge-augmented reasoning distillation, where, during training, the small LM learns to generate rationales given the training data and the retrieved knowledge by the rationale. (Right, § 4.2) Illustration of reranker training (top) and inference (bottom). Reranker learns to prioritize the passage which has knowledge relevant to the rationale.

Intuitively, the rationale provides a deeper and more comprehensive understanding of the reasoning behind the answer associated with the question, which better guides the small model to correctly answer the question [18]. Although previous works [33, 17, 38, 12, 18] have also leveraged the rationales generated by LLMs to make small models excel at diverse reasoning tasks, generating rationales for knowledge-intensive tasks with a small LM requires additional care. As previously described in Section 3, the reasoning distillation with a small model but without knowledge augmentation may degrade the quality of the rationale generation due to the incapability of memorizing training data with the small model [57, 27] and the necessity of the memorization for better performance in language tasks [4]. Therefore, the rationale generation should be evidenced by extrinsic knowledge from external memory to enhance the capability of the small LM for generating a high-quality rationale.

**Integrating External Knowledge Base** Motivated by Theorem 2, we propose to retrieve a passage from an external Knowledge Base (KB) which is a corpus of over millions of documents $\mathcal{D} = \{\boldsymbol{d}_1, \ldots, \boldsymbol{d}_K\}$ to support memorization capacity of the small LM. Note that the acquisition of the relevant document from KB is crucial for training the small LM to generate high-quality rationale which leads to correct answers for given questions. As done in open-domain QA task [7], we retrieve a set of relevant passages for a given query with the sparse retriever BM25 [46]. In order to obtain the document which is the most relevant to the rationale $\boldsymbol{r}_{ij}$ generated by the LLM, we utilize the rationale as a query to retrieve a set of passages $\hat{\mathcal{D}}_{ij} = \texttt{topk}(\rho(\boldsymbol{d}|\boldsymbol{r}_{ij}; \mathcal{D}), k) \subset \mathcal{D}$, where $\rho$ denotes a retriever scoring the document $\boldsymbol{d} \in \mathcal{D}$ based on relevance to the query $\boldsymbol{r}_{ij}$ and $\texttt{topk}$ yields the $k$ passages with the top-$k$ highest relevance scores. Finally, we utilize the retrieved documents $\hat{\mathcal{D}}_{ij}$ for fine-tuning the small LM to generate the rationale $\boldsymbol{r}_{ij}$ and answer $\boldsymbol{y}_i$ for the question $\boldsymbol{x}_i$ as follows:

$$\mathcal{L}_{\texttt{distill-KB}}(\theta) = -\frac{1}{n \cdot l} \sum_{i=1}^{n} \sum_{j=1}^{l} \log p_\theta(\boldsymbol{r}_{ij}, \boldsymbol{y}_i | \boldsymbol{x}_i, \hat{\mathcal{D}}_{ij}), \tag{2}$$

where the rationale and answer are sequentially generated as we did in Equation 1.

## 4.2 Training Neural Reranker for Rationale Generation

The remaining issue is that we cannot use the rationale as a query at the inference time. As an alternative, we can use the question $\boldsymbol{x}_i$ instead of the rationale $\boldsymbol{r}_{ij}$ as a query to retrieve a set of passages with the retriever. However, there is no guarantee that the top-$k$ passages retrieved by the input $\boldsymbol{x}_i$ as a query contain relevant information to generate correct rationales. In detail, based on the question as a query, the retriever can obtain a set of passages that contain relevant documents for generating rationales with a sufficiently large $k$ but $k \ll K$. However, the target documents we want for rationale generation may be assigned with low rankings and thus they may not be chosen for knowledge augmentation at the inference time. To remedy this issue, we propose to leverage a neural reranker $f_\phi$ [26] with parameter $\phi$ to re-rank the set of passages retrieved by the retriever $\rho$ so that we can acquire more relevant documents for generating rationale at the inference time.

In order to train the neural reranker, we might manually construct a ground truth passage for each question. However, we assume a realistic setting where the ground truth passage for reranker training

is not given. Instead, we train the reranker to imitate how the retriever scores the passage $\boldsymbol{d} \in \mathcal{D}$ with the rationale $\boldsymbol{r}_{ij}$ as a query. Specifically, we first utilize the retriever $\rho$ to obtain a set of passages from $\mathcal{D}$ with the rationale $\boldsymbol{r}_{ij}$ as a query as follows: $\tilde{\mathcal{D}}_{ij} = \texttt{topk}(\rho(\boldsymbol{d}|\boldsymbol{r}_{ij}; \mathcal{D}), \kappa_1) \bigcup \texttt{topk}(\rho(\boldsymbol{d}|\boldsymbol{x}_i; \mathcal{D}), \kappa_2)$ where $\kappa_1$ and $\kappa_2$ are the number of candidate documents (Figure 2 is the case where $\kappa_2 = 0$). Then, we normalize the score $\rho(\boldsymbol{d}|\boldsymbol{r}_{ij}; \mathcal{D})$ of the document from $\tilde{\mathcal{D}}_{ij}$, denoted as $Q(\boldsymbol{d}|\boldsymbol{r}_{ij})$. Similarly, we use the reranker $f_\phi$ to score each document in $\tilde{\mathcal{D}}_{ij}$ with the given question $\boldsymbol{x}_i$ and normalize the score denoted as $P_\phi(\boldsymbol{d}|\boldsymbol{x}_i)$. We use softmax for normalization with hyperparameters $\tau_1, \tau_2 > 0$ as follows:

$$Q(\boldsymbol{d}|\boldsymbol{r}_{ij}) = \frac{\exp\left(\rho(\boldsymbol{d}|\boldsymbol{r}_{ij}; \mathcal{D})/\tau_1\right)}{\sum_{\boldsymbol{d}' \in \tilde{\mathcal{D}}_{ij}} \exp\left(\rho(\boldsymbol{d}'|\boldsymbol{r}_{ij}; \mathcal{D})/\tau_1\right)}, \; P_\phi(\boldsymbol{d}|\boldsymbol{x}) = \frac{\exp(f_\phi(\boldsymbol{d}, \boldsymbol{x}_i)/\tau_2)}{\sum_{\boldsymbol{d}' \in \tilde{\mathcal{D}}_{ij}} \exp(f_\phi(\boldsymbol{d}', \boldsymbol{x}_i)/\tau_2)},$$

where $\boldsymbol{d} \in \tilde{\mathcal{D}}_{ij}$. Finally, we minimize the KL divergence between $Q(\boldsymbol{d}|\boldsymbol{r}_{ij})$ and $P_\phi(\boldsymbol{d}|\boldsymbol{x}_i)$:

$$\mathcal{L}_{\texttt{rerank}}(\phi) = \frac{1}{n \cdot l} \sum_{i=1}^{n} \sum_{j=1}^{l} D_{\mathrm{KL}}(Q(\boldsymbol{d}|\boldsymbol{r}_{ij}) \| P_\phi(\boldsymbol{d}|\boldsymbol{x}_i)). \tag{3}$$

Intuitively, the objective function guides the reranker to assign higher scores to passages that are similar to the rationale $\boldsymbol{r}_{ij}$. Note that both objective $\mathcal{L}_{\texttt{distill-KB}}(\theta)$ and $\mathcal{L}_{\texttt{rerank}}(\phi)$ are independent; therefore, we do not need to jointly update both of the small LM and the reranker.

### 4.3 Inference

After training, we obtain the small LM with the parameter $\theta^* \in \arg\min_\theta \mathcal{L}_{\texttt{distill-KB}}(\theta)$ and the reranker with the parameter $\phi^* \in \arg\min_\phi \mathcal{L}_{\texttt{rerank}}(\phi)$. At the test time, to answer the question $\boldsymbol{x}_*$, we first get a set of candidate documents $\tilde{\mathcal{D}}_* = \texttt{topk}(\rho(\boldsymbol{d}|\boldsymbol{x}_*; \mathcal{D}), \kappa^*)$ with the retriever $\rho$ and $\kappa^* = 100$. Then we re-rank all the document $d \in \tilde{\mathcal{D}}_*$ with $f_{\phi^*}$ and choose top-$k$ relevant documents w.r.t the question $\boldsymbol{x}_*$ as follows: $\hat{\mathcal{D}}_* = \texttt{topk}(\{f_{\phi^*}(\boldsymbol{d}, \boldsymbol{x}_*) \mid \boldsymbol{d} \in \tilde{\mathcal{D}}_*\}, k)$. Finally, we generate a rationale $\boldsymbol{r}_* = \arg\max_{\boldsymbol{r}} p_{\theta^*}(\boldsymbol{r}|\boldsymbol{x}_*, \hat{D}_*)$ and an answer $\boldsymbol{y}_* = \arg\max_{\boldsymbol{y}} p_{\theta^*}(\boldsymbol{y}|\boldsymbol{r}_*, \boldsymbol{x}_*, \hat{\mathcal{D}}_*)$.

## 5 Experiments

### 5.1 Experimental Setting

**Task and Dataset** In our experiments, we focus on knowledge-intensive reasoning tasks which require both the reasoning ability over the knowledge and the compound knowledge of the specific domain. As our primary benchmark, we use the medical multiple-choice question dataset — **MedQA-USMLE** [23]. The dataset contains 12,723 4-option multiple-choice question answering problems from US medical licensing exam. This dataset is the best fit to evaluate our method since 98% of the questions simulate the realistic clinical settings by presenting patient cases that require extensive professional domain-specific knowledge and complex reasoning ability over multiple evidence sources. To further validate our approach, we employ **StrategyQA** [14] dataset, which involves 2,780 yes/no questions that demand sophisticated multi-step reasoning skills and the ability to gather supporting evidence from various domains. We additionally validate our approach on commonsense reasoning with **OpenbookQA** [39] dataset, which consists of 5,957 elementary-level science questions with 4 multiple-choice options.

**Baselines** We compare our method against relevant baselines. **Few-shot In-context Learning** (ICL) utilizes a few training samples as a prompt to make a prediction [5]. **Few-shot ICL + Chain-of-Thought** (CoT) leverages chain-of-thought prompting to generate a rationale and generate an answer based on the rationale [28]. **Fine-tuning** refers to the one that fine-tunes a pre-trained model to generate an answer given only a question. The performance of the above baselines represents the capability of a small language model to solve knowledge-intensive reasoning tasks using only training data but without any extrinsic guidance on reasoning or external knowledge.

To assess the impact of external knowledge, we augment the above three baselines with documents retrieved from the knowledge base (Wikipedia), denoted as **Knowledge-Augmented** models. For the knowledge augmentation, we append retrieved passages along with the question at both training and inference time. These baselines help us understand how much external knowledge can improve the performance of each baseline. We also compare our KARD against the standard **Reasoning Distillation** without knowledge-augmentation [33, 17, 38, 12, 18].

Table 1: Experimental results on the **MedQA-USMLE** dataset with Flan-T5 [53] and OPT [20, 59] models. We report the mean and standard deviation of accuracy with 3 different runs for reasoning distillation methods.

| Method | MedQA-USMLE (Flan-T5 [53]) | | | MedQA-USMLE (OPT [20, 59]) | |
| --- | --- | --- | --- | --- | --- |
| | Base (250M) | Large (780M) | XL (3B) | 350M | 1.3B-IML |
| Few-shot | 23.49 | 31.50 | 35.66 | 27.42 | 29.14 |
| Few-shot + Chain-of-Thought (CoT) | 25.22 | 32.21 | 32.99 | 25.06 | 26.39 |
| *Knowledge-Augmented* Few-shot + CoT | 31.34 | 32.60 | 34.41 | 25.84 | 28.75 |
| Fine-tuning | 30.71 | 34.49 | 37.39 | 26.47 | 25.77 |
| *Knowledge-Augmented* Fine-tuning | 33.39 | 37.71 | 39.12 | 25.84 | 28.67 |
| Reasoning Distillation | $31.03_{\pm.40}$ | $39.62_{\pm.29}$ | $46.32_{\pm.36}$ | $29.43_{\pm1.13}$ | $34.30_{\pm.95}$ |
| KARD *(ours, BM25)* | $33.14_{\pm.23}$ | $41.87_{\pm.93}$ | $47.27_{\pm.67}$ | $30.79_{\pm.78}$ | $35.48_{\pm.37}$ |
| KARD *(ours, Reranker)* | $\mathbf{38.15}_{\pm.39}$ | $\mathbf{44.59}_{\pm.47}$ | $\mathbf{48.94}_{\pm.32}$ | $\mathbf{32.86}_{\pm1.12}$ | $\mathbf{38.83}_{\pm.46}$ |
| KARD *(Silver Knowledge, Oracle)* | 40.30 | 49.80 | 53.50 | 35.90 | 42.18 |
| CoT from ChatGPT *(Teacher, Oracle)* | 61.59 | 65.51 | 67.16 | - | 50.27 |

Table 2: Experimental results on the **StrategyQA** and **OpenbookQA** dataset with T5 models [45]. † indicates experiments with Flan-T5 having the same size. We report experimental results as in Table 1.

| Method | StrategyQA (T5 [45]) | | | OpenbookQA (T5 [45]) | | |
| --- | --- | --- | --- | --- | --- | --- |
| | Base | Large | XL | Base | Large | XL |
| Few-shot | 48.47 | 48.47 | 51.67 | 23.00 | 27.60 | 25.00 |
| Few-shot + CoT | 48.47 | 48.33 | 48.76 | 27.60 | 27.40 | 27.80 |
| *KA* Few-shot + CoT | 48.47 | 48.91 | 48.76 | 27.60 | 27.60 | 27.80 |
| Fine-tuning | 52.26 | 56.33 | 51.53 | 54.00 | 62.00 | 74.60 |
| *KA* Fine-tuning | 52.11 | 58.81 | 53.38 | 53.80 | 64.60 | 73.80 |
| Reasoning Distillation | $55.36_{\pm.27}$ | $64.97_{\pm.55}$ | $68.41_{\pm.48}$ | $58.87_{\pm.50}$ | $66.13_{\pm.34}$ | $77.00_{\pm.59}$ |
| KARD *(ours, BM25)* | $55.90_{\pm.24}$ | $65.94_{\pm.12}$ | $68.8_{\pm1.08}$ | $55.93_{\pm.38}$ | $64.40_{\pm.71}$ | $76.00_{\pm.28}$ |
| KARD *(ours, Reranker)* | $\mathbf{56.57}_{\pm.25}$ | $\mathbf{66.04}_{\pm.60}$ | $\mathbf{70.55}_{\pm.81}$ | $\mathbf{59.33}_{\pm.74}$ | $\mathbf{66.40}_{\pm.16}$ | $\mathbf{78.53}_{\pm.25}$ |
| KARD *(Silver Kn., Oracle)* | 57.50 | 65.65 | 72.34 | 63.40 | 72.40 | 82.00 |
| CoT from ChatGPT *(Oracle)*† | 66.38 | 67.10 | 72.05 | 58.60 | 78.80 | 87.80 |

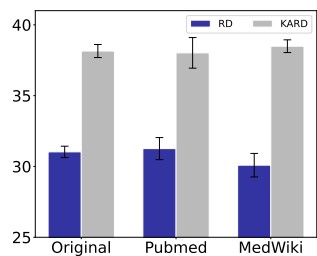

Figure 3: Experimental results on MedQA-USMLE, where we perform DAPT [15] on each dataset of x-axis with Flan-T5 Base.

As **oracle** models, we present a variant of KARD that receives better knowledge as input. In particular, at the inference time, we augment KARD with the silver document which is the passage retrieved with the gold rationale generated by the LLM as a query. This model represents an upper bound of the neural reranker performance. Additionally, we directly provide the small instruction fine-tuned language models (Flan-T5 [53] and OPT-IML [20]) with the rationale from the LLM in inference, to assess the upper bound of the performance gain on small models with high-quality rationales.

**Language Models**    For all the experiments, we use the T5 models [45] including Flan-T5 [9], and OPT models [59] including OPT-IML [20]. For the reranker, we use LinkBERT models [56]. As for the teacher LLM, we employ `GPT-3.5-turbo` (ChatGPT) [42] through the proprietary API.

See **Appendix B** for experimental settings in detail.

### 5.2   Experimental Results

Table 1 shows that KARD consistently outperforms all the baselines on the MedQA-USMLE dataset on both encoder-decoder (Flan-T5) and decoder-only (OPT) language models. Remarkably, KARD exhibits a substantial positive effect on smaller models, as evident from the significant performance gain of the Flan-T5 Base model, which has 250 million parameters, over a fine-tuning baseline on the MedQA-USMLE dataset. Regarding the analysis of the model size, please refer to Section 5.3. The impact of KARD decreases as the size of the model increases since larger models can better memorize knowledge during pre-training and fine-tuning. Moreover, we empirically show that knowledge augmentation consistently improves performance not only in reasoning distillation but also in few-shot chain-of-thought and fine-tuning. It is worth noting that this empirical evidence supports our theoretical analysis in Section 3 that knowledge augmentation enhances the performance of small models. Furthermore, our experimental results indicate that the reranker consistently improves the performance of models for all sizes, over the retrieval with BM25. From the experimental results with silver knowledge (oracle), there is room for improvement by retrieving more relevant documents, which can help the model generate a high-quality rationale.

We also present additional experimental results on StrategyQA and OpenbookQA datasets in Table 2. Once again, KARD outperforms all baselines in experiments with both datasets. Notably, compared to MedQA-USMLE, the few-shot methods on StrategyQA and OpenbookQA exhibit performance similar to random guessing, as T5 lacks the ability of in-context learning [9]. Furthermore, fine-tuning

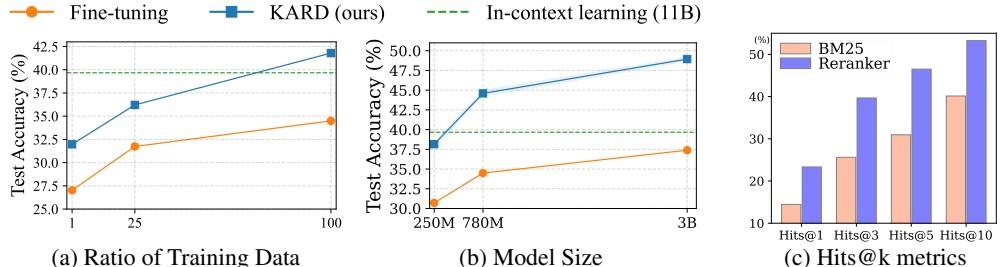

(a) Ratio of Training Data      (b) Model Size      (c) Hits@k metrics

Figure 4: **(a) Efficiency on training data and (b) model size.** On MedQA-USMLE, we compare KARD against the fine-tuning baseline by varying either the number of training data with Flan-T5 Large or the number of parameters, including the few-shot in-context learning performance of Flan-T5 XXL (11B). **(c)** Considering silver documents as ground truth, we measure **Hits@k** on the documents retrieved by BM25 and the reranker.

Table 3: Analysis on rationale diversity.      Table 4: Analysis on $\kappa^*$.      Table 5: Analysis on $k$.

| | **BM25** | |
|---|---|---|
| Rationales | Base | Large |
| $l = 3$ | 30.09 | 35.43 |
| $l = 5$ | 32.13 | 39.04 |
| $l = 10$ | 32.91 | 41.79 |

| | **Reranker** | |
|---|---|---|
| Passages | Base | Large |
| $\kappa^* = 20$ | 36.45 | 43.91 |
| $\kappa^* = 50$ | 36.06 | 44.23 |
| $\kappa^* = 100$ | 36.76 | 45.25 |

| | **Flan-T5 Base** | |
|---|---|---|
| Passages | BM25 | Reranker |
| $k = 1$ | 32.91 | 36.76 |
| $k = 2$ | 32.84 | 37.71 |
| $k = 3$ | 32.36 | 37.39 |

T5-XL on StrategyQA results in poor performance since it fails to generalize to the test data. On the other hand, reasoning distillation improves the performance of models across all different sizes on both datasets. Our KARD further yields performance improvement over the reasoning distillation baseline, demonstrating the effectiveness of knowledge augmentation in both datasets.

### 5.3 Analysis

**Experiments with DAPT** Domain-Adaptive Pre-Training (DAPT) [15] is the useful strategy to adapt Pre-trained Language Models (PLMs) on the specific domain to effectively tackle the tasks on it, which is done by further pre-training the PLM on a large-scale domain-specific text corpus [2, 30, 37]. As it is interesting to observe whether the DAPT can enhance the capacity of PLMs for reasoning distillation in domain-specific knowledge-intensive tasks by further performing training on relevant domain-specific data before distillation, we conduct experiments with models from DAPT. Specifically, we further pre-train the Flan-T5 Base model on two moderate-scale biomedical corpora, Pubmed abstracts and MedWiki [35], respectively. Then, we apply reasoning distillation and KARD to PLMs with further pre-trained parameters. In Figure 3, we observe that DAPT on Pubmed marginally enhances the performance of reasoning distillation. On the other hand, KARD contributes more substantially to performance improvement than DAPT. This result indicates that KARD offers a distinct advantage in knowledge-intensive reasoning tasks compared to DAPT.

**Efficiency on Dataset and Model Sizes** To validate the efficiency of our KARD in terms of training data and model size, we measure the test accuracy on the MedQA-USMLE dataset while varying the number of training data and model parameters. As shown in Figure 4a, our KARD can effectively transfer the reasoning ability of the LLM with the proposed KARD mechanism, using only a small number of training data. Moreover, the gaps between the naive fine-tuning and our KARD become much larger when increasing the number of training data, which confirms that we can potentially increase the effectiveness of KARD with more training data for knowledge-augmented distillation from LLMs. Furthermore, it is worth noting that KARD is a *sample-efficient*. With 25% of the training data, KARD outperforms the same model fine-tuned on the full data.

For the efficiency in terms of the model size, as shown in Figure 4b, KARD with 250M parameters achieves higher accuracy than the fine-tuned model with 3B parameters (14 times larger). Moreover, KARD with 780M parameters outperforms the 11B in-context learning baseline. These results show the significant practical advantage of our KARD in resource-restricted settings since the small LM with KARD requires significantly less computational cost yet it outperforms the LLM.

**Retrieval Performance** To evaluate the performance of the reranker on MedQA-USMLE, we consider the top-3 silver documents retrieved with the rationales generated by LLM as the ground

Table 6: **Rationale generation example** of KARD, compared with Reasoning Distillation in MedQA-USMLE with Flan-T5 XL. The answer is in blue. We mark the hallucination in the generated rationale as yellow and highlight the factually correct statements in the generated rationale and its supporting fact in the passage as red.

**Question**: A 29-year-old woman presents to the clinic after several months of weight loss. She noticed a 6.8 kg (15 lb) unintentional weight loss over the preceding several months. She has not changed her diet or exercise habits. She also reports feuding with her boyfriend over the temperature of their shared apartment, as she always feels warmer than he does. The vital signs include: heart rate 110/min and blood pressure 146/78 mm Hg. The physical exam is notable for warm and slightly moist skin. Which of the following is the best single treatment option for this patient? (A) Glucocorticoids **(B) Methimazole** (C) Propranolol (D) Radioiodine therapy

| | Retrieved Passage | Outputs |
|---|---|---|
| **Reasoning Distillation** *(without knowledge)* | *No knowledge* | The patient's symptoms are consistent with hyperthyroidism, specifically Graves' disease. Glucocorticoids are the first-line treatment for Graves' disease. Methimazole, propranolol, and radioiodine therapy can be used as adjunctive treatment. Answer: A |
| **KARD** *(Ours, reranker)* | Hyperthyroidism . (...) The best single treatment option for Graves' disease is methimazole, which may temporarily help people while other treatments are having effect. Glucocorticoids and radioiodine therapy can also be used, but they are not the best option in this case. Propranolol can help control the heart rate and blood pressure, but it will not treat (...) | The symptoms and physical exam findings suggest hyperthyroidism, specifically Graves' disease. The best single treatment option for Graves' disease is methimazole, which works by inhibiting the thyroid hormone synthesis. Glucocorticoids and radioiodine therapy can also be used, but they are not the best option. Answer: B |

truth, and measure Hits@k on the documents retrieved by BM25 and reranker with $\kappa^* = 100$. In Figure 4c, the reranker achieves significantly better Hits@k than BM25. This result indicates that the reranker successfully learns to prioritize passages that are helpful to generate correct rationale at the test time, which leads to performance improvement on the knowledge-intensive reasoning tasks.

**The Number of Rationales During Training**   Following Ho et al. [17], we generate multiple rationales for each training sample in order to facilitate diverse reasoning in small language model training. In Table 3, we present the impact of rationale diversity during training on both Flan-T5 base and large models using the MedQA-USMLE dataset. As the number of rationales per training data increases, the performance also improves, demonstrating the benefit of employing multiple rationales. However, the performance gains become small when we increase the number of rationals from 5 to 10. This suggests that utilizing more diverse rationales beyond 10 may not yield significant further improvements, at least in the MedQA-USMLE dataset.

**The Number of Candidate Documents for Reranker**   It is crucial to determine the size of the candidate document set ($\kappa^*$) to which the reranker assigns the relevance scores w.r.t a question. In Table 4, we present the performance of both Flan-T5 base and large models on MedQA-USMLE, while varying $\kappa^*$. The results indicate that increasing the number of candidate documents tends to be beneficial, as it allows the reranker to consider a broader range of diverse candidate documents.

**The Number of Passages Used for Inference**   Even LLMs tend to be easily distracted by irrelevant context [47]. Therefore, simply adding more passages during inference may not necessarily enhance performance if relevant knowledge is not selected. In Table 5, we present the impact of the number of passages used in KARD during inference ($k$ in Section 4.3) on Flan-T5 Base and MedQA-USMLE. We observe that the performance of KARD (BM25) without the re-ranker decreases with increasing $k$. This result implies that using additional passages does not always result in generating better rationales. In contrast, using two passages ($k = 2$) with the reranker is better than a single passage ($k = 1$). This result indicates that the reranker effectively selects more suitable knowledge than BM25, thereby contributing to performance improvement in the MedQA-USMLE benchmark.

**Qualitative Analysis**   In Table 6, we provide an example comparing the rationale generated by our KARD against the rationale by the baseline model with reasoning distillation but without knowledge augmentation. We choose one sample from the MedQA-USMLE dataset and generate the rationale using the Flan-T5 XL model. The model without knowledge augmentation generates the rationale that seems plausible based on the given instruction and question. However, it mistakenly generates the hallucination that Glucocorticoids are the first-line treatment for Graves' disease, which is incorrect. As a result, it fails to predict the correct answer (B) Methimazole. In contrast, when the model is fine-tuned with KARD, it generates a correct rationale that is supported by the retrieved knowledge indicating that methimazole is the best single treatment option for Graves' disease. Consequently, it successfully predicts the correct answer. This example highlights the effectiveness of our KARD method for generating accurate rationales by incorporating relevant knowledge, which leads to an improved question answering performance on knowledge-intensive reasoning benchmarks.

# 6 Discussion

## 6.1 Comparison to Retrieval-augmented Generation

Retrieval-Augmented Generation (RAG) [32] primarily focuses on solving knowledge-intensive tasks (e.g., open-domain QA), where the accurate knowledge retrieval is important to achieve higher performance. In terms of the methodology, the key differences between KARD and RAG are that RAG utilizes the question as a query and jointly fine-tunes the generator and retriever. To quantitatively analyze the advantage of our KARD against RAG in reasoning distillation, we conduct experiments with RAG on the reasoning distillation with two datasets that we used in the main experiment, where we use Flan-T5 base for MedQA-USMLE and T5 base for StrategyQA as base LMs and DPR [25] as the trainable retriever for RAG. In Table 7, experimental results show that using RAG in reasoning distillation achieves lower accuracy than KARD, showing that our KARD is more tailored approach to reasoning distillations.

Table 7: Experimental results including RAG on Reasoning Distillation (RD) with (Flan-)T5 base.

|  | MedQA | StrategyQA |
|---|---|---|
| *KA* Fine-tuning | 33.39 | 52.11 |
| RAG + RD | 24.84 | 54.24 |
| KARD *(Reranker)* | **38.15** | **56.57** |

## 6.2 Failure Case Analysis

In Table 1, we can see significant differences between KARD with reranker on the Flan-T5 XL and the ChatGPT in MedQA-USMLE. Our investigation focuses on understanding the cause of these gaps by examining samples where our method fails while ChatGPT succeeds. We collect 30 samples from corresponding cases and categorize them into two groups. The first group consists of cases where the reranker fails to obtain the document relevant to generating the correct rationale. The second group includes cases where the small language model fails to produce correct rationales and makes incorrect predictions, despite having access to relevant knowledge in the retrieved document. Out of 30 samples, 15 fall into the first category, while the remaining 15 belong to the second category. This observation indicates the need for further improvements in both retriever and distillation methods to enhance the performance of small language models in knowledge-intensive reasoning tasks.

## 6.3 Limitations

We have shown substantial improvements in small LMs' performance on knowledge-intensive reasoning tasks through our KARD. However, it is important to acknowledge the limitations of our study. First, in terms of methodology, the effectiveness of our knowledge augmentation heavily relies on the quality of the document retrieved from the external knowledge base. As indicated in Table 1 and Figure 4c, our reranker substantially improves the performance of small models by retrieving better knowledge. Despite the diminishing performance gap between BM25 and the reranker as the model size increases, there is still a significant difference between the document retrieved by the reranker and the silver knowledge. This indicates that the reranker might miss important passages that can augment the knowledge of even large LMs. Therefore, further advancements in retrieval methods are necessary to generate better rationale, as this remains an important research challenge even for large language models [13]. Second, regarding experiments, we have tested our approach on relatively small LMs having under 3B parameters given our limited computational budgets. However, exploring the use of relatively larger language models like GPT-3 [5, 44] or LLaMA [50, 51] with KARD could be of great interests, which is a promising direction for future research.

# 7 Conclusion

In this work, we proposed Knowledge-Augmented Reasoning Distillation (KARD) which enhances the capabilities of small Language Models (LMs) on knowledge-intensive reasoning tasks that demand both knowledge and reasoning abilities. Our approach involves generating rationales from large LMs and fine-tuning small LMs on these rationales, while augmenting small LMs with external knowledge from a non-parametric memory. Our theoretical analysis motivates our method by demonstrating the effectiveness of external memory in reducing the memorization requirements of small LMs. Through empirical experiments, we showed that KARD outperforms traditional approaches such as fine-tuning and reasoning distillation, thereby providing a pathway to improve small LMs in reasoning tasks that require a comprehensive understanding of domain-specific knowledge.

## Acknowledgements

This work was done while the first author was working at AITRICS. We would like to thank Kangwook Lee and the anonymous reviewers for their insightful comments and suggestions regarding this work, which helped us to make improvements to the paper. This work was supported by AITRICS, the Institute of Information & communications Technology Planning & Evaluation (IITP) grant funded by the Korea government (MSIT) (No. 2019-0-00075, Artificial Intelligence Graduate School Program (KAIST), No. RS-2022-00187238, Development of Large Korean Language Model Technology for Efficient Pre-training, and No.2022-0-0071), the National Research Foundation of Korea (NRF) grant funded by the Korea government (MSIT) (No. RS-2023-00256259 and NRF-2018R1A5A1059921) and Samsung Electronics (IO201214-08145-01).

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

# Appendix

## A    Motivation: Effect of Knowledge-Augmentation on Memorization

### A.1    Additional Details and Discussion

We adopted the abstracted language problem, i.e., the next-symbol prediction problem with reference strings $c_j \sim \text{Uniform}\left(\{0,1\}^d\right)$, considered in the main text of [4] with no symbol corruption. In this problem, the random process $P \sim q$ is defined by drawing $c_j \in \{0,1\}^d$ uniformly at random as $c_j \sim \text{Uniform}\left(\{0,1\}^d\right)$ for all $j \in [N]$: a set $\{c_j\}_{j=1}^N$ corresponds to a $P$. Then, the random process $(Z,Y) \sim P$ is defined by sampling $j \sim \text{Uniform}\left([N]\right)$ and $\ell \sim \text{Uniform}\left(\{0,1,2,\ldots,d-1\}\right)$ and by setting $Z = (j, c_j(1:\ell))$ and $Y = c_j(\ell+1)$, where $c_j(1:l)$ denotes the first $l$ symbols of $c_j$ and $c_j(l+1)$ is $(l+1)$-th symbol of $c_j$. The training data $X \sim P^{\otimes n}$ contains $((Z_i, Y_i))_{i=1}^n$ generated by the same process $n$ times independently.

Recall that the inference algorithm $\varphi$ uses a KB denoted by $S$ such that $|S| = N+R$ and $\{c_j\}_{j=1}^N \subseteq S$. Here, $S$ is a set and *not* ordered; therefore we do not know the identifier $j$ of each $s \in S$ and which $s \in S$ is useful given each $(Z,Y)$. Thus, this still requires learning and memorizing some information from the training data $X$.

In Theorem 2, $N$ (or $R$) is the number of useful (or unuseful) documents in KB to be extracted by a retriever. Thus, $N+R$ is the total number of possible documents to be retrieved. As $N+R$ increases, we need to memorize more information of the training data to get the best match at test time. Theorem 2 shows that this increase happens only in the log rate, $\log_2(N+R)$. Thus, we can get improvements when the size of KB ($d$) is larger than *the log* of the number of possible choices to be retrieved by the retriever ($\log_2(N+R)$).

Moreover, unlike Theorem 1, we can remove the dependence on $n$ and have $I(X; \mathcal{A}(X)|P) = O(N\log_2(N+R))$ in Theorem 2 when the training size $n$ grows at a faster rate than the number of the useful documents $N$. This is because our proof does not rely on the high probability of having a singleton training sample per each reference $c_j$ unlike the proof of [4].

### A.2    Proof of Theorem 2

*Proof.* Let $\epsilon > 0$ and $m = \log_2((1 - (\frac{N-1}{N})^n)\frac{(N+R)^2 - (N+R)}{2\epsilon})$. Since the distribution over subpopulation is uniform,

$$\text{err}_{q,n}^\varphi(\mathcal{A}) = \sum_{j=1}^N \Pr(Q_j)\Pr(E_0|Q_j) = \frac{1}{N}\sum_{j=1}^N \Pr(E_0|Q_j),$$

where $E_0$ is the event of $\varphi(Z, \mathcal{A}(X), S) \neq Y$ and $Q_j$ is the event that the subpopulation of the test sample $(Z,Y) \sim P$ is $j$. Let $E_1^j$ be the event of having at least one training data sample $((Z_i, Y_i) \in X)$ in the subpopulation $j$ of the test sample $(Z,Y) \sim P$. Then,

$$\Pr(E_0|Q_j) = \Pr(E_1^j|Q_j)\Pr(E_0|Q_j, E_1^j) + \Pr(\bar{E}_1^j|Q_j)\Pr(E_0|Q_j, \bar{E}_1^j)$$

where $\bar{E}_1^j$ is the complement of $E_1^j$. Denote by $\ell_x^j$ the length of training sample, and by $\ell_t^j$ the length of the test sample, in the subpopulation $j$. Let $E_2^j$ be the event of $\ell_x^j \geq m$ for at least one training sample in the subpopulation $j$, and $E_3^j$ be the event of $\ell_t^j < \ell_x^j$ for at least one training sample in the subpopulation $j$. Then,

$$\begin{aligned}
\text{err}_{q,n}^\varphi(\mathcal{A}) =& \frac{1}{N}\sum_{j=1}^N \Pr(E_1^j|Q_j)\Pr(E_2^j|Q_j, E_1^j)\Pr(E_0|Q_j, E_1^j, E_2^j) \\
&+ \frac{1}{N}\sum_{j=1}^N \Pr(E_1^j|Q_j)\Pr(\bar{E}_2^j|Q_j, E_1^j)\Pr(E_3^j|Q_j, E_1^j, \bar{E}_2^j)\Pr(E_0|Q_j, E_1^j, \bar{E}_2^j, E_3^j) \\
&+ \frac{1}{N}\sum_{j=1}^N \Pr(E_1^j|Q_j)\Pr(\bar{E}_2^j|Q_j, E_1^j)\Pr(\bar{E}_3^j|Q_j, E_1^j, \bar{E}_2^j)\Pr(E_0|Q_j, E_1^j, \bar{E}_2^j, \bar{E}_3^j)
\end{aligned}$$

$$+ \frac{1}{N} \sum_{j=1}^{N} \Pr(\bar{E}_1^j | Q_j) \Pr(E_0 | Q_j, \bar{E}_1^j).$$

Define $E_0^*$ to be the event of $M_{\mathrm{OPT}}(Z) \neq Y$ where $M_{\mathrm{OPT}} = \mathcal{A}_{\mathrm{OPT}}(X)$. Then, the same decomposition holds for $\mathrm{err}_{q,n}(\mathcal{A}_{\mathrm{OPT}})$ yielding that

$$\mathrm{err}_{q,n}(\mathcal{A}_{\mathrm{OPT}}) = \frac{1}{N} \sum_{j=1}^{N} \Pr(E_1^j | Q_j) \Pr(E_2^j | Q_j, E_1^j) \Pr(E_0^* | Q_j, E_1^j, E_2^j)$$

$$+ \frac{1}{N} \sum_{j=1}^{N} \Pr(E_1^j | Q_j) \Pr(\bar{E}_2^j | Q_j, E_1^j) \Pr(E_3^j | Q_j, E_1^j, \bar{E}_2^j) \Pr(E_0^* | Q_j, E_1^j, \bar{E}_2^j, E_3^j)$$

$$+ \frac{1}{N} \sum_{j=1}^{N} \Pr(E_1^j | Q_j) \Pr(\bar{E}_2^j | Q_j, E_1^j) \Pr(\bar{E}_3^j | Q_j, E_1^j, \bar{E}_2^j) \Pr(E_0^* | Q_j, E_1^j, \bar{E}_2^j, \bar{E}_3^j)$$

$$+ \frac{1}{N} \sum_{j=1}^{N} \Pr(\bar{E}_1^j | Q_j) \Pr(E_0^* | Q_j, \bar{E}_1^j).$$

Since the probabilities are non-negative, by ignoring some terms, we have that

$$\mathrm{err}_{q,n}(\mathcal{A}_{\mathrm{OPT}}) \geq \frac{1}{N} \sum_{j=1}^{N} \Pr(\bar{E}_1^j | Q_j) \Pr(E_0^* | Q_j, \bar{E}_1^j)$$

$$+ \frac{1}{N} \sum_{j=1}^{N} \Pr(E_1^j | Q_j) \Pr(\bar{E}_2^j | Q_j, E_1^j) \Pr(\bar{E}_3^j | Q_j, E_1^j, \bar{E}_2^j) \Pr(E_0^* | Q_j, E_1^j, \bar{E}_2^j, \bar{E}_3^j).$$

Since these two terms $\Pr(E_0^* | Q_j, \bar{E}_1^j)$ and $\Pr(E_0^* | Q_j, E_1^j, \bar{E}_2^j, \bar{E}_3^j)$ correspond to the errors when the test data comes into the place with no information from the training data, the memorizing data does not help and the random guess achieves the best performance. Moreover, these two cases can be detected at test time by $(\varphi, \mathcal{A})$ by memorizing the first $\min(m, \ell_x^j)$ bits of an input sequence in the training data $X$ and $j$ (i.e., $\min(m, \ell_x^j) + 1$ bits in total) for at most one training sample per $j \in [N]$. Thus, we choose $\mathcal{A}$ to memorize the first $\min(m, \ell_x^j)$ bits and $j$ for at most one training sample per $j \in [N]$; i.e., we memorize $\min(m, \ell_x^j) + 1 \leq m + 1$ bits for at most $\min(N, n)$ training samples. In total, we memorise at most $\min(N, n)(m + 1)$ bits. Then, these two terms are the same for $\mathrm{err}_{q,n}(\mathcal{A}_{\mathrm{OPT}})$ and $\mathrm{err}_{q,n}^\varphi(\mathcal{A})$ by detecting these cases (with memorization) and generating a random guess for these cases. Thus,

$$\mathrm{err}_{q,n}(\mathcal{A}_{\mathrm{OPT}}) \geq \frac{1}{N} \sum_{j=1}^{N} \Pr(\bar{E}_1^j | Q_j) \Pr(E_0 | Q_j, \bar{E}_1^j)$$

$$+ \frac{1}{N} \sum_{j=1}^{N} \Pr(E_1^j | Q_j) \Pr(\bar{E}_2^j | Q_j, E_1^j) \Pr(\bar{E}_3^j | Q_j, E_1^j, \bar{E}_2^j) \Pr(E_0 | Q_j, E_1^j, \bar{E}_2^j, \bar{E}_3^j).$$

Therefore,

$$\mathrm{err}_{q,n}^\varphi(\mathcal{A}) - \mathrm{err}_{q,n}(\mathcal{A}_{\mathrm{OPT}})$$

$$\leq \frac{1}{N} \sum_{j=1}^{N} \Pr(E_1^j | Q_j) \Pr(E_2^j | Q_j, E_1^j) \Pr(E_0 | Q_j, E_1^j, E_2^j)$$

$$+ \frac{1}{N} \sum_{j=1}^{N} \Pr(E_1^j | Q_j) \Pr(\bar{E}_2^j | Q_j, E_1^j) \Pr(E_3^j | Q_j, E_1^j, \bar{E}_2^j) \Pr(E_0 | Q_j, E_1^j, \bar{E}_2^j, E_3^j)$$

$$+ \frac{1}{N} \sum_{j=1}^{N} \Pr(E_1^j | Q_j) \Pr(\bar{E}_2^j | Q_j, E_1^j) \Pr(\bar{E}_3^j | Q_j, E_1^j, \bar{E}_2^j) \Pr(E_0 | Q_j, E_1^j, \bar{E}_2^j, \bar{E}_3^j)$$

$$+ \frac{1}{N} \sum_{j=1}^{N} \Pr(\bar{E}_1^j|Q_j) \Pr(E_0|Q_j, \bar{E}_1^j) - \frac{1}{N} \sum_{j=1}^{N} \Pr(\bar{E}_1^j|Q_j) \Pr(E_0|Q_j, \bar{E}_1^j)$$

$$- \frac{1}{N} \sum_{j=1}^{N} \Pr(E_1^j|Q_j) \Pr(\bar{E}_2^j|Q_j, E_1^j) \Pr(\bar{E}_3^j|Q_j, E_1^j, \bar{E}_2^j) \Pr(E_0|Q_j, E_1^j, \bar{E}_2^j, \bar{E}_3^j).$$

$$= \frac{1}{N} \sum_{j=1}^{N} \Pr(E_1^j|Q_j) \Pr(E_2^j|Q_j, E_1^j) \Pr(E_0|Q_j, E_1^j, E_2^j)$$

$$+ \frac{1}{N} \sum_{j=1}^{N} \Pr(E_1^j|Q_j) \Pr(\bar{E}_2^j|Q_j, E_1^j) \Pr(E_3^j|Q_j, E_1^j, \bar{E}_2^j) \Pr(E_0|Q_j, E_1^j, \bar{E}_2^j, E_3^j)$$

Here, we notice that $\Pr(E_0|Q_j, E_1^j, \bar{E}_2^j, E_3^j) = 0$ because $\mathcal{A}$ memorizes the first $\min(m, \ell_x^j)$ bits, where $\min(m, \ell_x^j) = \ell_x^j$ conditioned on $\bar{E}_2^j$, and because this is conditioned on the events $E_1^j$ and $E_3^j$. That is, $\Pr(E_0|Q_j, E_1^j, \bar{E}_2^j, E_3^j)$ is the error probability when we have a training sample with the length $\ell_x^j$ larger than the test sample length $\ell_t^j$ on the same subpopulation and $\ell_x^j = \min(m, \ell_x^j)$ bits of the training samples are memorized. In this case, we can simply output the next symbol for the test sample from the memorized training sample, yielding $\Pr(E_0|Q_j, E_1^j, \bar{E}_2^j, E_3^j) = 0$. Therefore,

$$\mathrm{err}_{q,n}^\varphi(\mathcal{A}) - \mathrm{err}_{q,n}(\mathcal{A}_{\mathrm{OPT}}) \leq \frac{1}{N} \sum_{j=1}^{N} \Pr(E_1^j|Q_j) \Pr(E_2^j|Q_j, E_1^j) \Pr(E_0|Q_j, E_1^j, E_2^j). \quad (4)$$

In the case of the events $Q_j, E_1^j$ and $E_2^j$, $\mathcal{A}(X)$ memorized $\min(m, \ell_x^j) = m$ bits of the input sequence in the training sample for at least one training sample in the subpopulation $j$ of the test sample. In this case, at test time, $\varphi$ first chooses one training sample in the same subpopulation $j$ of the test sample (we can do this because the non-label part $Z$ contains the subpopulation information $j$ in both test sample $(Z, Y) \sim P$ and in the training sample; see Brown et al. [4]). Then, $\varphi$ uses the memorized $m$ bits of this training sample to search $S$; i.e., it picks $s \in S$ such that the first $m$ bits of $s$ matches with the memorized $m$ bits of the training sample. If there are more than one $s \in S$ that satisfies this condition, then it randomly picks one of them. Note that in this case of the events $Q_j, E_1$ and $E_2$, there is at least one match; there exists $s \in S$ the first $m$ bits of $s$ matches with the memorized $m$ bits of the training sample.

Thus, if there is no more than one $s \in S$ that satisfies this condition, then the inference algorithm $\varphi$ uniquely determines the reference string $c_j$ of the test sample $(Z, Y) \sim P$. With the reference string $c_j$ of the test sample, $\varphi$ then predict the next symbol of the test sample by outputting $(\ell_t^j + 1)$-th symbol of the reference string $c_j$, where $\ell_t^j$ is computable by computing the length of the test input $Z$ without any information of training samples. Therefore, if there is no more than one $s \in S$ that satisfies the above condition, then $\varphi$ makes a correct prediction. This implies that if $s$ and $s'$ do *not* have the same first $m$ bits for all $s, s' \in S$ with $s \neq s'$, then $\varphi$ makes a correct prediction. By taking complement, this means that the error occurs only if there exist $s, s' \in S$ with $s \neq s'$ such that $s$ and $s'$ have the same first $m$ bits. Therefore,

$$\Pr(E_0|Q_j, E_1^j, E_2^j) \leq \Pr(\exists s, s' \in S \text{ with } s \neq s' \text{ such that } s \text{ and } s' \text{ have the same first } m \text{ bits})$$

$$= \Pr(\cup_{\{s,s'\} \in \mathcal{T}} \{s \text{ and } s' \text{ have the same first } m \text{ bits}\})$$

$$\leq \sum_{\{s,s'\} \in \mathcal{T}} \Pr(\{s \text{ and } s' \text{ have the same first } m \text{ bits}\})$$

$$\leq \binom{N+R}{2} \frac{1}{2^m} = \frac{(N+R)^2 - (N+R)}{2} \frac{1}{2^m}$$

where $\mathcal{T} = \{\{s, s'\}|s, s' \in S, s \neq s'\}$. Since $m = \log_2((1 - (\frac{N-1}{N})^n)\frac{(N+R)^2-(N+R)}{2\epsilon})$,

$$\Pr(E_0|Q_j, E_1^j, E_2^j) \leq \frac{(N+R)^2 - (N+R)}{2} \frac{1}{(1 - (\frac{N-1}{N})^n)\frac{(N+R)^2-(N+R)}{2\epsilon}}$$

$$= \frac{\epsilon}{1 - (\frac{N-1}{N})^n}.$$

Plugging this into equation 4,

$$\text{err}_{q,n}^{\varphi}(\mathcal{A}) - \text{err}_{q,n}(\mathcal{A}_{\text{OPT}}) \leq \frac{1}{N} \sum_{j=1}^{N} \Pr(E_1^j | Q_j) \Pr(E_2^j | Q_j, E_1^j) \frac{\epsilon}{1 - (\frac{N-1}{N})^n}.$$

Since $\Pr(E_2^j | Q_j, E_1^j) \leq 1$ and $\Pr(E_1^j | Q_j) = 1 - \Pr(\bar{E}_1^j | Q_j^j) = 1 - (\frac{N-1}{N})^n$,

$$\text{err}_{q,n}^{\varphi}(\mathcal{A}) - \text{err}_{q,n}(\mathcal{A}_{\text{OPT}}) \leq \frac{1}{N} \sum_{j=1}^{N} \epsilon = \epsilon.$$

Since we only require $m + 1$ bits for at most $\min(N, n)$ training samples from the above construction of $\mathcal{A}$ and $\varphi$, we have that $I(X; \mathcal{A}(X)|P) = O(\min(N, n)m)$ while achieving $\text{err}_{q,n}^{\varphi}(\mathcal{A}) - \text{err}_{q,n}(\mathcal{A}_{\text{OPT}}) \leq \epsilon$ for any $\epsilon > 0$. □

# B Implementation Details

**Rationale Generation** As for the teacher LLM, we employ `GPT-3.5-turbo` (ChatGPT) [42] through the public API. We demonstrate the prompt we used for rationale generation in Table 11, 12, and 13. Specifically, we utilize the instruction and 5-shot examples from Singhal et al. [48] for MedQA-USMLE, and the chain-of-thought prompt Kojima et al. [28] for StrategyQA and Openbook-QA. We generate multiple $l$ rationales for each training sample with the LLM. This allows training the small LM with a diverse set of rationales. Furthermore, we utilize the filtering method [17] to remove incorrect rationales from the training set but we use a small Flan-T5 base to filter the rationales which lead to incorrect prediction.

**Training** For all our experiments, we fine-tune the small language model for 3 epochs with a batch size of 32 using the AdamW optimizer [36] and a learning rate of $10^{-4}$. Each model utilizes a maximum of 96GB GPU memory with 4 NVIDIA TITAN RTX GPUs for fine-tuning. In the StrategyQA and OpenbookQA experiments, we use the T5 model instead of Flan-T5 to prevent any potential data contamination with the corresponding test set, as Flan-T5 is fine-tuned on both datasets during instruction tuning. For the number of documents used for knowledge augmentation during KARD training, we set $k = 1$ for MedQA-USMLE and StrategyQA and $k = 3$ for OpenbookQA; specifically, we append documents retrieved from the retriever $\rho$ along with each training sample to construct the input for training. See Tables 14 and 15 for examples of the input and output used in KARD training. For the train-test split of dataset, we use the official split for MedQA-USMLE [23] and OpenbookQA [39]. For strategyQA, we split the training set into $7 : 3$ ratio to build the in-house test set following Ho et al. [17].

**Inference** When it comes to methods that require rationale generation, such as Chain-of-Thought and reasoning distillation, we employ a technique called self-consistency [52] during inference. Specifically, for each question, a model generates multiple rationales and corresponding predictions during the inference, followed by a majority voting to choose the final answer among the predictions.

**Retriever** We use Wikipedia as the external knowledge base for both all of datasets. For the retriever $\rho$, we use BM25 [46] which is the term-frequency-based sparse retrieval method. To implement BM25, we use the `pyserini` library[2] which provides a reproducible information retrieval framework.

**Reranker** To implement a neural reranker, we adopt the scoring method used in ColBERT [26]. We use BioLinkBERT-base and LinkBERT-base [56] as the backbone language model for MedQA-USMLE and StrategyQA, respectively. For reranker training, we utilize LoRA [19] for efficient training. For hyperparameters, we set $\tau_1 = 1$ and $\tau_2 = 100$. For all datasets, we train the reranker for 3 epochs with a batch size of 16 using AdamW optimizer and a learning rate of $10^{-4}$. We set $\kappa_1 = 8, \kappa_2 = 0$ for MedQA-USMLE and $\kappa_1 = 4, \kappa_2 = 4$ for StrategyQA and OpenbookQA.

---

[2]https://github.com/castorini/pyserini

Table 8: Analysis on retriever.

| Retriever | Flan-T5 Base | |
|---|---|---|
| | Wikipedia | Pubmed |
| BM25 | 33.14 | 31.58 |
| DPR | 29.77 | - |
| Reranker | **38.15** | **36.84** |
| Silver (oracle) | 40.30 | 45.48 |

Table 9: Analysis on $\kappa_1$ and $\kappa_2$.

| $\kappa_1$ | $\kappa_2$ | **Flan-T5 Base** |
|---|---|---|
| $\kappa_1 = 4$ | $\kappa_2 = 4$ | 36.84 |
| $\kappa_1 = 0$ | $\kappa_2 = 8$ | 34.96 |
| $\kappa_1 = 8$ | $\kappa_2 = 0$ | 37.71 |

Table 10: Analysis on queries.

| Query | Flan-T5 Base | |
|---|---|---|
| | BM25 | Reranker |
| $\boldsymbol{x}_i$ | 31.81 | 35.51 |
| $\boldsymbol{r}_{ij}$ | 32.91 | 37.71 |

**DAPT** In Section 5.3 **Experiments with DAPT**, we conduct domain-adaptive pre-training [15] to assess the impact of further pre-training prior to the reasoning distillation. We further pre-train Flan-T5 base model on the LM objective as discussed in the original T5 and prompt-tuning paper [45, 31]. We use two different corpus for this experiment. One is the PubMed abstracts[3] which contains the 1.8 abstracts extracted from the biomedical papers uploaded in PubMed. Another is MedWiki which is the subset of the Wikipedia passages containing the biomedical knowledge [35]. We further train the model for 3 epochs with a batch size of 128 using the AdamW optimizer and a learning rate of $10^{-4}$.

## C   More Analysis

### C.1   Retriever and Knowledge Base

In our main experiments, we utilize BM25 [46] as the retriever and Wikipedia as the knowledge base. Regarding these choices, several questions arise: (1) Does using a dense retriever provide more advantages than a sparse retriever? (2) Does utilizing a domain-specific knowledge base offer more benefits than a general knowledge base?

To address these questions, we perform an analysis on the choice of retriever and knowledge base. First, we substitute the external knowledge base with Pubmed abstracts corpus[4], which consists of abstracts from publicly available medical papers containing specialized knowledge. On the other hand, we maintain the knowledge base as Wikipedia but replace the BM25 retriever with DPR [25], one of the popular dense retrievers in open-domain question answering tasks.

In Table 8, we present the results of our analysis. We observe that the use of PubMed does not result in performance improvement compared to Wikipedia. We speculate that incorporating a reranker with PubMed corpus could potentially yield further improvements considering that using silver knowledge from PubMed yields superior performance against using silver knowledge from Wikipedia; however, we leave it as a future work. Furthermore, using DPR rather leads to degraded performance. We hypothesize that DPR may struggle to generalize to our specific task since it is primarily designed for open-domain QA tasks. It is plausible that employing a more adaptable retriever such as contriever [21] might offer greater benefits, but we also leave this as an avenue for future research.

### C.2   $\kappa_1$ and $\kappa_2$ for Reranker Training

In the training of the reranker (as described in Section 4.2 of the main paper), the reranker learns to prioritize passages that are relevant to the rationale. We normalize the score $\rho(\boldsymbol{d}|\boldsymbol{r}_{ij};\mathcal{D})$ of the passages in the candidate set through softmax. Therefore, the reranker $f_\phi$ learns a relative score $f_\phi(\boldsymbol{d}, \boldsymbol{x}_i)$ between passages in the candidate set during training. Due to computational constraints, the number of passages in the candidate set is limited. As a result, it is necessary to determine which passages should be included in each candidate set $\tilde{\mathcal{D}}_{ij}$.

There are two potential sources for the candidate passages. One approach is to retrieve candidate passages using the rationale $\boldsymbol{r}_{ij}$ as a query, while another approach is to use the question (input data) $\boldsymbol{x}_i$ as the query. It is intuitive to consider combining candidate passages from both approaches, as this can provide additional information to the reranker. Specifically, the reranker can learn the differences between passages that are highly similar to the question and passages that are highly similar to the

---
[3]https://huggingface.co/datasets/ywchoi/pubmed_abstract_0
[4]https://pubmed.ncbi.nlm.nih.gov/

rationale. This training setup enables the reranker to prioritize passages that are more relevant to the rationale among the passages retrieved from the question as a query during inference time.

In the implementation, we can control the composition of the candidate set by adjusting hyperparameters $\kappa_1$ and $\kappa_2$ while fixing the candidate size as 8. In Table 9, we provide an analysis of the impact of both values on the reranker performance, by measuring the task accuracy with the Flan-T5 base model. We observe that relying solely on passages related to the question for composing the candidate set is not a viable approach, as it hinders the reranker's ability to learn which passage is relevant to the rationale effectively.

### C.3 Knowledge used in KARD training ($\hat{\mathcal{D}}_{ij}$)

In order to enhance the reasoning ability of a small LM, it is crucial to retrieve passages that contain the proper knowledge to assist the small LM in generating rationales obtained from the large language model. It is intuitive to utilize the rationale itself as a query for retrieval, focusing to retrieve passages that are relevant to the given rationale. To validate this intuition empirically, we instead use the question as a query for the retrieval during training, retrieving passages denoted as $\texttt{topk}(\rho(\boldsymbol{d}|\boldsymbol{x}_i; \mathcal{D}), k)$. In Table 10, we present the empirical evidence that using the passages relevant to the question during training actually leads to performance degradation, as these passages are highly unlikely to contain the necessary knowledge for generating rationales.

### C.4 Examples of Failure Cases

In Section 6.1 of the main paper, we provide an analysis of the failure cases. In this section, we showcase an example for each category of failure cases.

The first category involves the case where the reranker fails to retrieve pertinent knowledge required for generating a rationale. In Table 16, we present an example of the failure case that corresponds to the first category. In this example, the reranker fails to retrieve the relevant passage necessary for generating the correct rationale. Therefore, the small Language Model (LM) generates rationales that include a hallucination: suggesting warm compresses as the most suitable treatment for allergic conjunctivitis, which is factually incorrect.

The second category is the case where the reranker successfully retrieves the relevant passage which is the silver passage obtained from the gold rationale from ChatGPT. In the example illustrated in Table 17, the retrieved passage contains information about pulmonary contusion, which is one of the answer options. If the small LM comprehends the given passage accurately, it should recognize that pulmonary contusion is not the correct answer, as elevated pulmonary capillary wedge pressure and troponins are unrelated to pulmonary contusion but rather associated with a cardiac contusion. However, despite being trained with KARD, the small LM fails to reason based on this knowledge, resulting in the generation of the incorrect rationale.

Furthermore, we also provide examples of failure cases from StrategyQA. Unlike MedQA-USMLE, most of the failure cases in StrateqyQA fall into the first category, where the reranker fails. This is due to the fact that StrategyQA questions often require complex and multiple sources of knowledge to answer.

In Table 18, we present an example that corresponds to the first category. The question requires knowledge about the background color of the Azerbaijani flag as well as the color composition of the Powerpuff Girls. However, the reranker retrieves a passage related to the flag of Japan, which is not helpful for answering the given question.

In Table 19, we show the case that belongs to the second category, which is a rare case in StrategyQA. In this example, the reranker successfully retrieves a passage containing information about the number of fused vertebrae in human sacrums. However, the small LM fails to answer the question due to its inability to comprehend the retrieved passage. Moreover, since the question requires additional knowledge about the number of fused vertebrae in the sacrums of an Alaskan Malamute, it is challenging to answer the question solely based on the available knowledge, particularly if the small LM lacks intrinsic knowledge regarding this specific domain.

# D  Broader Impact

Our proposed approach aims to enhance the performance of small language models in tasks that involve knowledge-intensive reasoning. As demonstrated in Section 5 of the main paper, our method is beneficial in domains that require professional knowledge, such as the medical field.

However, it is crucial to exercise caution when employing small language models with our method in real-world clinical applications. These models have the potential to generate statements that are factually incorrect, as explicitly mentioned in Section 6 of the main paper and Section C.4. Therefore, thorough attention and careful consideration are required when utilizing small language models in such contexts, even with our proposed method.

It is worth noting that even large language models, which have not been extensively validated in real-world clinical settings, should also be used with caution. The potential for generating inaccurate information exists across various language models, and their deployment in clinical sites should be approached with careful consideration and validation [48, 41, 40].

Table 11: MedQA [23]. 5-shot Chain-of-Thought prompt [48] for rationale generation with ChatGPT [42].

The following are multiple-choice questions about medical knowledge. Generate a detailed step-by-step explanation for each question and answer.

**Question:** A 22-year-old male marathon runner presents to the office with the complaint of right-sided rib pain when he runs long distances. Physical examination reveals normal heart and lung findings and an exhalation dysfunction at ribs 4-5 on the right. Which of the following muscles or muscle groups will be most useful in correcting this dysfunction utilizing a direct method?
(A) anterior scalene (B) latissimus dorsi (C) pectoralis minor (D) quadratus lumborum
**Answer:** (C)
Explanation: We refer to Wikipedia articles on medicine for help. Among the options, only pectoralis minor muscle origins from the outer surfaces of the 3rd to 5th ribs.

**Question:** A 36-year-old male presents to the office with a 3-week history of low back pain. He denies any recent trauma but says that he climbs in and out of his truck numerous times a day for his job. Examination of the patient in the prone position reveals a deep sacral sulcus on the left, a posterior inferior lateral angle on the right, and a lumbosacral junction that springs freely on compression. The most likely diagnosis is
(A) left-on-left sacral torsion (B) left-on-right sacral torsion (C) right unilateral sacral flexion (D) right-on-right sacral torsion
**Answer:** (D)
**Explanation:** We refer to Wikipedia articles on medicine for help. The deep sulcus on the left, a posterior ILA on the right, with a negative spring test suggests a right-on-right sacral torsion. All other options have a deep sulcus on the right.

**Question:** A 44-year-old man comes to the office because of a 3-day history of sore throat, nonproductive cough, runny nose, and frontal headache. He says the headache is worse in the morning and ibuprofen does provide some relief. He has not had shortness of breath. Medical history is unremarkable. He takes no medications other than the ibuprofen for pain. Vital signs are temperature 37.4°C (99.4°F), pulse 88/min, respirations 18/min, and blood pressure 120/84 mm Hg. Examination of the nares shows erythematous mucous membranes. Examination of the throat shows erythema and follicular lymphoid hyperplasia on the posterior oropharynx. There is no palpable cervical adenopathy. Lungs are clear to auscultation. Which of the following is the most likely cause of this patient's symptoms?
(A) Allergic rhinitis (B) Epstein-Barr virus (C) Mycoplasma pneumonia (D) Rhinovirus
**Answer:** (D)
**Explanation:** We refer to Wikipedia articles on medicine for help. The symptoms, especially the headache, suggest that the most likely cause is Rhinovirus. Epstein-Barr virus will cause swollen lymph nodes but there is no palpable cervical adenopathy. Lungs are clear to auscultation suggests it's not Mycoplasma pneumonia.

**Question:** A previously healthy 32-year-old woman comes to the physician 8 months after her husband was killed in a car crash. Since that time, she has had a decreased appetite and difficulty falling asleep. She states that she is often sad and cries frequently. She has been rechecking the door lock five times before leaving her house and has to count exactly five pieces of toilet paper before she uses it. She says that she has always been a perfectionist but these urges and rituals are new. Pharmacotherapy should be targeted to which of the following neurotransmitters?
(A) Dopamine (B) Glutamate (C) Norepinephrine (D) Serotonin
**Answer:** (D)
**Explanation:** We refer to Wikipedia articles on medicine for help. The patient feels sad and among the options, only Dopamine and Serotonin can help increase positive emotions. Serotonin also affects digestion and metabolism, which can help the patient's decreased appetite and sleep difficulty.

**Question:** A 42-year-old man comes to the office for preoperative evaluation prior to undergoing adrenalectomy scheduled in 2 weeks. One month ago, he received care in the emergency department for pain over his right flank following a motor vehicle collision. At that time, blood pressure was 160/100 mm Hg and CT scan of the abdomen showed an incidental 10-cm left adrenal mass. Results of laboratory studies, including complete blood count, serum electrolyte concentrations, and liver function tests, were within the reference ranges. The patient otherwise had been healthy and had never been told that he had elevated blood pressure. He takes no medications. A follow-up visit in the office 2 weeks ago disclosed elevated urinary normetanephrine and metanephrine and plasma aldosterone concentrations. The patient was referred to a surgeon, who recommended the adrenalectomy. Today, vital signs are temperature 36.6°C (97.9°F), pulse 100/min, respirations 14/min, and blood pressure 170/95 mm Hg. Physical examination discloses no significant findings. Initial preoperative preparation should include treatment with which of the following?
(A) Labetalol (B) A loading dose of potassium chloride (C) Nifedipine (D) Phenoxybenzamine
**Answer:** (D)
**Explanation:** We refer to Wikipedia articles on medicine for help. The symptoms and the adrenal mass suggested pheochromocytoma, and the blood pressure indicates hypertension. Phenoxybenzamine is used to treat hypertension caused by pheochromocytoma.

**Question:** [question] **Answer:** [answer] **Explanation:**

Table 12: StrategyQA [14]. 0-shot Chain-of-Thought prompt [28] for rationale generation with ChatGPT [42].

The following are multiple choice questions (with answers). Generate a detailed step-by-step explanation for each question and answer.

**Question:** [question]
**Answer:** [answer]
**Explanation:** Let's think step by step.

---

Table 13: OpenbookQA [39]. 3-shot Chain-of-Thought prompt for rationale generation with ChatGPT [42].

The following are multiple choice questions (with answers). Generate a detailed step-by-step explanations for each question and answer.

**Question:** The sun is responsible for
(A) puppies learning new tricks (B) children growing up and getting old (C) flowers wilting in a vase (D) plants sprouting, blooming and wilting
**Answer:** (D)
Explanation: We refer to basic knowledge about the effects of sun on living organisms. The sun provides the energy required for photosynthesis in plants, which allows them to sprout, bloom, and eventually wilt. The other options are not related to the effects of the sun.

**Question:** When standing miles away from Mount Rushmore
(A) the mountains seem very close (B) the mountains are boring (C) the mountains look the same as from up close (D) the mountains seem smaller than in photographs
**Answer:** (D)
**Explanation:** This question requires some basic knowledge about perspective and the way our eyes perceive distance. When we stand miles away from an object like Mount Rushmore, it appears smaller than it does in photographs because our eyes are capturing a smaller visual angle. Option (A) is incorrect because if the mountains seemed very close, we would not be standing miles away. Option (B) is subjective and not related to perceptual phenomena. Option (C) is untrue because when we are up close to Mount Rushmore, we can see details that we cannot see from far away.

**Question:** When food is reduced in the stomach
(A) the mind needs time to digest (B) take a second to digest what I said (C) nutrients are being deconstructed (D) reader's digest is a body of works
**Answer:** (C)
**Explanation:** This question requires basic knowledge about the digestive system. When food is in the stomach, it is broken down and deconstructed into nutrients that can be absorbed by the body. Option (A) is incorrect because the mind is not directly involved in the digestive process. Option (B) is a play on words and not related to digestion. Option (D) is a reference to a literary magazine and not related to the digestive process.

**Question:** [question] **Answer:** [answer] **Explanation:**

Table 14: Input and output example for KARD (training) in MedQA-USMLE [23].

---

**INPUT:**

The following are multiple-choice questions about medical knowledge. Generate a step-by-step explanation for each question:

**Question:** A 23-year-old pregnant woman at 22 weeks gestation presents with burning upon urination. She states it started 1 day ago and has been worsening despite drinking more water and taking cranberry extract. She otherwise feels well and is followed by a doctor for her pregnancy. Her temperature is 97.7°F (36.5°C), blood pressure is 122/77 mmHg, pulse is 80/min, respirations are 19/min, and oxygen saturation is 98% on room air. Physical exam is notable for an absence of costovertebral angle tenderness and a gravid uterus. Which of the following is the best treatment for this patient?

A. Ampicillin B. Ceftriaxone C. Doxycycline D. Nitrofurantoin

**Knowledge:** Urinary tract infection . Urinary tract infections are more concerning in pregnancy due to the increased risk of kidney infections. During pregnancy, high progesterone levels elevate the risk of decreased muscle tone of the ureters and bladder, which leads to a greater likelihood of reflux, where urine flows back up the ureters and towards the kidneys. While pregnant women do not have an increased risk of asymptomatic bacteriuria, if bacteriuria is present they do have a 25–40% risk of a kidney infection. Thus if urine testing shows signs of an infection—even in the absence of symptoms—treatment is recommended. Cephalexin or nitrofurantoin are typically used because they are generally considered safe in pregnancy. A kidney infection during pregnancy may result in premature birth or pre-eclampsia (a state of high blood pressure and kidney dysfunction during pregnancy that can lead to seizures). Some women have UTIs that keep coming back in pregnancy and currently there is not enough research on how to best treat these infections.

**Explanation:**

**OUTPUT:**

We refer to Wikipedia articles on medicine for help. The patient is pregnant, which limits the options for treatment due to the risk of harm to the fetus. Nitrofurantoin is considered safe for use in pregnancy and is an effective treatment for uncomplicated urinary tract infections. Ampicillin and Ceftriaxone are not typically used to treat UTIs, and Doxycycline is contraindicated in pregnancy.

**Answer:** D

---

Table 15: Input and output example for KARD (training) in StrategyQA [14].

---

**INPUT:**

The following are multiple-choice questions. Generate a step-by-step explanation for each question:

**Question:** Is the cuisine of Hawaii suitable for a vegan?

A. yes B. no

**Knowledge:** Cuisine of the Mariana Islands . The cuisine of the Mariana Islands is largely made up of meat dishes including pork, poultry, horse meat, and meat from other land mammals. Some popular land animals consumed include bats, mice, rats and lizards. Most of the dishes on the islands are cooked in coconut oil. Naturally, the islands' surrounding waters make seafood another popular option. Some seafoods include sea cucumbers, whale's meat, and various fish. It is said that the Mariana Island's cuisine is heavily influenced by its neighbors Papuan, Hawaiian, and American cuisines.The Mariana's cuisine is very international as well. The people enjoy many Asian dishes such as kimchi.

**Explanation:**

**OUTPUT:**

Hawaiian cuisine typically includes a lot of seafood, pork, and other types of meat. These ingredients are not suitable for a vegan as they come from animals. While there may be some plant-based options in Hawaiian cuisine like fruits, vegetables, and grains, the majority of dishes are not vegan-friendly. Therefore, the correct answer is (B) no.

**Answer:** B

---

Table 16: **MedQA-USMLE Example of Failure Case** *(first category)*. We compare the generated output from the Flan-T5 XL model with KARD and reranker against the gold correct rationale from ChatGPT. The answer is in blue. We mark the hallucination in the generated rationale as yellow and highlight the factually correct statements in the generated rationale and its supporting fact in the passage as red.

**Question**: A 35-year-old man comes to the physician because of itchy, watery eyes for the past week. He has also been sneezing multiple times a day during this period. He had a similar episode 1 year ago around springtime. He has iron deficiency anemia and ankylosing spondylitis. Current medications include ferrous sulfate, artificial tear drops, and indomethacin. He works as an elementary school teacher. His vital signs are within normal limits. Visual acuity is 20/20 without correction. Physical examination shows bilateral conjunctival injection with watery discharge. The pupils are 3 mm, equal, and reactive to light. Examination of the anterior chamber of the eye is unremarkable. Which of the following is the most appropriate treatment? (A) Erythromycin ointment **(B) Ketotifen eye drops** (C) Warm compresses (D) Fluorometholone eye drops

| | Retrieved Passage | Outputs |
|---|---|---|
| **KARD** *(Ours, reranker)* | Mansonella perstans . A 36-year-old man was admitted to the outpatient clinic at the Goundi Missionary Hospital in the south of Chad in May 2001. He complained of visual impairment in the left eye, ocular and abdominal pruritus, and abdominal pain. He had previously been treated with DEC for "M. perstans" infection five months prior to his visit. A blood sample was taken at 11:00 am, and examined microscopically as a thick blood film stained with Giemsa's solution. The thick blood film revealed the presence of "M. perstans", and no other parasites were found. He had 3% eosinophilia. A visual acuity test showed a reduction of visual acuity to 4/10 for the left eye, while the right eye was 9/10. (...) | We refer to Wikipedia articles on medicine for help. The symptoms suggest allergic conjunctivitis, and the most appropriate treatment is warm compresses for comfort and relief. Erythromycin ointment is used for bacterial conjunctivitis, ketotifen eye drops are used for glaucoma, and fluorometholone eye drops are used for dry eyes. Answer: C |
| **Gold** *(Oracle)* | Emedastine . Emedastine (trade name Emadine) is a second generation antihistamine used in eye drops to alleviate the symptoms of allergic conjunctivitis. It acts as a H receptor antagonist. It works by blocking the action of histamine that causes allergic symptoms. It is used in form of the difumarate. The emedastine difumarate is a white, crystalline, water-soluble fine powder. Emedastine eye drops is usually applied twice a day to the affected eye. When the patients with allergic conjunctivitis were treated with 0.05% emedastine difumarate ophthalmic solution for six weeks, the signs and symptoms such as redness, itching and swelling of the eyes were relieved. Emedastine appears to be devoid of effects on adrenergic, dopaminergic and serotonin receptors. This drug was developed by Alcon, which is global medical company specializing in eye care products. | We refer to Wikipedia articles on medicine for help. The patient's symptoms suggest allergic conjunctivitis, which can be treated with antihistamine eye drops like Ketotifen. Erythromycin ointment is used to treat bacterial infections, warm compresses may help with symptoms but do not treat the underlying issue, and Fluorometholone eye drops are used to treat inflammation and do not address the allergic response. Answer: B |

Table 17: **MedQA-USMLE Example of Failure Case** *(second category)*. We compare the generated output from the Flan-T5 XL model with KARD and reranker against the gold correct rationale from ChatGPT. The answer is in blue. We mark the hallucination in the generated rationale as yellow and highlight the factually correct statements in the generated rationale and its supporting fact in the passage as red.

**Question**: A 27-year-old man presents to the emergency department after a motor vehicle collision. The patient was the front seat unrestrained driver in a head on collision. The patient's echocardiogram (ECG) is notable only for sinus tachycardia. His temperature is 99.5°F (37.5°C), blood pressure is 107/58 mmHg, pulse is 120/min, respirations are 17/min, and oxygen saturation is 98% on room air. The patient is given 2 liters of Ringer lactate solution and morphine. Initial workup demonstrates that the patient's pulmonary capillary wedge pressure and troponins are elevated. The patient is currently complaining of chest pain. Physical exam is notable for an uncomfortable young man with bruising over his chest wall. Which of the following is the most likely diagnosis? **(A) Cardiac contusion** (B) Hemorrhage (C) Pulmonary contusion (D) Takotsubo cardiomyopathy

| | Retrieved Passage | Outputs |
|---|---|---|
| **KARD** *(Ours, reranker)* | Pulmonary contusion . A pulmonary contusion, also known as lung contusion, is a bruise of the lung, caused by chest trauma. As a result of damage to capillaries, blood and other fluids accumulate in the lung tissue. The excess fluid interferes with gas exchange, potentially leading to inadequate oxygen levels (hypoxia). Unlike pulmonary laceration, another type of lung injury, pulmonary contusion does not involve a cut or tear of the lung tissue. A pulmonary contusion is usually caused directly by blunt trauma but can also result from explosion injuries or a shock wave associated with penetrating trauma. With the use of explosives during World Wars I and II, pulmonary contusion resulting from blasts gained recognition. In the 1960s its occurrence in civilians began to receive wider recognition, in which cases it is usually caused by traffic accidents. The use of seat belts and airbags reduces the risk to vehicle occupants. Diagnosis is made by studying the cause of the injury, physical examination and chest radiography. Typical signs and symptoms include direct effects of the physical trauma, such as chest pain and coughing up blood, as well as signs that the body is not receiving enough oxygen, such as cyanosis. The contusion frequently heals on its own with supportive care. | We refer to Wikipedia articles on medicine for help. The patient's history of front seat unrestrained driver in a motor vehicle collision and elevated pulmonary capillary wedge pressure and troponins suggest a pulmonary contusion as the most likely diagnosis. Cardiac contusion or hemorrhage would have presented with additional symptoms such as ECG changes and ECG changes in a different pattern, while Takotsubo cardiomyopathy is not typically associated with chest trauma or ECG changes. Answer: C |
| **Gold** *(Oracle)* | Pulmonary contusion . A pulmonary contusion, also known as lung contusion, is a bruise of the lung, caused by chest trauma. As a result of damage to capillaries, blood and other fluids accumulate in the lung tissue. The excess fluid interferes with gas exchange, potentially leading to inadequate oxygen levels (hypoxia). Unlike pulmonary laceration, another type of lung injury, pulmonary contusion does not involve a cut or tear of the lung tissue. A pulmonary contusion is usually caused directly by blunt trauma but can also result from explosion injuries or a shock wave associated with penetrating trauma. With the use of explosives during World Wars I and II, pulmonary contusion resulting from blasts gained recognition. In the 1960s its occurrence in civilians began to receive wider recognition, in which cases it is usually caused by traffic accidents. The use of seat belts and airbags reduces the risk to vehicle occupants. Diagnosis is made by studying the cause of the injury, physical examination and chest radiography. Typical signs and symptoms include direct effects of the physical trauma, such as chest pain and coughing up blood, as well as signs that the body is not receiving enough oxygen, such as cyanosis. The contusion frequently heals on its own with supportive care. | We refer to Wikipedia articles on medicine for help. The patient's elevated troponins and pulmonary capillary wedge pressure, along with chest pain and bruising over the chest wall, suggests cardiac contusion, which can occur from blunt chest trauma like in a motor vehicle collision. Hemorrhage would not cause elevated troponins and pulmonary capillary wedge pressure. Pulmonary contusion would cause respiratory symptoms. Takotsubo cardiomyopathy is a possibility, but not the most likely diagnosis given the patient's history of trauma. Answer: A |

Table 18: **StrategyQA Example of Failure Case** *(first category)*. We compare the generated output from the Flan-T5 XL model with KARD and reranker against the gold correct rationale from ChatGPT. The answer is in blue. We mark the hallucination in the generated rationale as yellow and highlight the factually correct statements in the generated rationale and its supporting fact in the passage as red.

| **Question**: Could the Powepuff Girls make the background to the Azerbaijani flag? **(A) yes** (B) no | | |
|---|---|---|
| | **Retrieved Passage** | **Outputs** |
| **KARD** *(Ours, reranker)* | Flag of Japan . Two recently designed national flags resemble the Japanese flag. In 1971, Bangladesh gained independence from Pakistan, and it adopted a national flag that had a green background, charged with an off-centered red disc that contained a golden map of Bangladesh. The current flag, adopted in 1972, dropped the golden map and kept everything else. The Government of Bangladesh officially calls the red disc a circle; the red color symbolizes the blood that was shed to create their country. The island nation of Palau uses a flag of similar design, but the color scheme is completely different. While the Government of Palau does not cite the Japanese flag as an influence on their national flag, Japan did administer Palau from 1914 until 1944. (...) | First, we need to know what the background of the Azerbaijani flag is. The flag of Azerbaijan has a blue background with white text and blue vertical bands. Second, we need to know what the Powepuff Girls look like. As far as we know, they are fictional character created by the fictional network Cartoon Network. Based on these two pieces of information, we can say that the Powepuff Girls are not a real country, and they do not have the capability or knowledge to design a flag background. The colors and design of the Azerbaijani flag are unique and original, and there is no evidence that they have ever been used as a background for a flag. Therefore, the answer to the question is (B) no, the Powepuff Girls could not make the background to the Azerbaijani flag. Answer: B |
| **Gold** *(Oracle)* | Flag of Azerbaijan . The national flag of Azerbaijan consists of three horizontal stripes of equal width, from top to bottom: light blue, red, and green. In the center are a white crescent and eight-pointed star. The basic description of the flag, along with the ratio, can be found in the Constitution of Azerbaijan, which was adopted 12 November 1995: | 1. Who are the Powerpuff Girls? They are three characters from an animated television series who have superpowers and fight crime. 2. What is the Azerbaijani flag? The flag has three horizontal stripes: blue, red, and green. In the center, there is a white crescent and an eight-pointed star. 3. Can the Powerpuff Girls make the background to the Azerbaijani flag? The answer is yes. The background consists of three horizontal stripes, which the Powerpuff Girls can easily create with their powers. They can make the stripes blue, red, and green in the correct order. It is not stated in the question whether they can create the crescent and star in the center, but since the question only asks about the background, we can assume that they do not need to create the symbols in the center. Therefore, the correct answer is (A) yes. Answer: A |

Table 19: **StrategyQA Example of Failure Case** *(second category).* We compare the generated output from the Flan-T5 XL model with KARD and reranker against the gold rationale from ChatGPT. The answer is in blue. We mark the hallucination in the generated rationale as yellow and highlight the factually correct statements in the generated rationale and its supporting fact in the passage as red.

| **Question**: Do human sacrums have more fused vertebrae than an Alaskan Malamute? (A) yes (B) no | | |
|---|---|---|
| | **Retrieved Passage** | **Outputs** |
| **KARD** *(Ours, reranker)* | Sacrum . The sacrum ( or ; plural: "sacra" or "sacrums";) in human anatomy is a large, triangular bone at the base of the spine, that forms by the fusing of sacral vertebrae S1-S5 between 18 and 30 years of age. | First, we need to understand what the sacrum is and how it is related to fused vertebrae. The sacrum is a triangular bone located at the base of the spine in humans. Next, we need to understand how vertebrae are fused. In the human spine, there are six fused vertebrae (S1, S2, S3, S4), while in the Alaskan Malamute, there are only two fused vertebrae (S3-S4). Now, we can compare the fused vertebrae in the human sacrum with those of the Alaskan Malamute. The human sacrum has six fused vertebrae, while the Alaskan Malamute has five fused vertebrae. Therefore, the answer to the question is (B) no, human sacrums do not have more fused vertebrae than an Alaskan Malamute. Answer: B |
| **Gold** *(Oracle)* | Axial skeleton . At birth the majority of humans have 33 separate vertebrae. However, during normal development several vertebrae fuse together, leaving a total of 24, in most cases. The confusion about whether or not there are 32-34 vertebrae stems from the fact that the two lowest vertebrae, the sacrum and the coccyx, are single bones made up of several smaller bones which have fused together. This is how the vertebrae are counted: 24 separate vertebrae and the sacrum, formed from 5 fused vertebrae and the coccyx, formed from 3-5 fused vertebrae. If you count the coccyx and sacrum each as one vertebra, then there are 26 vertebrae. If the fused vertebrae are all counted separately, then the total number of vertebrae comes to between 32 and 34. | Firstly, what is a sacrum? A sacrum is a triangular bone at the base of the spine that is composed of five fused vertebrae in humans and many other mammals. Secondly, what is an Alaskan Malamute? It is a breed of dog that is known for its strength and endurance. Now, the question is asking whether the human sacrums have more fused vertebrae than an Alaskan Malamute. In general, dog breeds tend to have more vertebrae than humans. However, when it comes to the specific case of the sacrum, the answer is that humans have more fused vertebrae. As mentioned earlier, human sacrums consist of five fused vertebrae, whereas Alaskan Malamutes have four. Therefore, the correct answer is (A) Yes. Answer: A |

