# OpenReview forum: "Knowledge-Augmented Reasoning Distillation for Small Language Models in Knowledge-Intensive Tasks"
_NeurIPS.cc/2023/Conference — NeurIPS 2023 poster_

### Official Review · Reviewer_Z8sP · 2023-07-04

**Soundness:** 3 good
**Presentation:** 4 excellent
**Contribution:** 3 good
**Rating:** 6
**Confidence:** 5

**Summary:**

The authors propose a form of knowledge distillation for a retriever-reader architecture. It uses rationales to guide the neural reranker to retrieve more relevant passages for reasoning, instead of passing the query to the retriever and retrieving the most similar passages. The paper includes an interesting set of ablations as well as some quantitative analysis and limitations. The generative process is as follows: at training time, a rationale is generated using a LLM. Successively, the rationale is passed to the retriever and it retrieves the top-k passages that are most similar to it Finally, a small LM is fine-tuned with the rationale, retrieved passages and question.

At inference time, the question is passed to the retriever, and top-k passages relevant to the question are retrieved (with BM25 plus a neural reranker which helps correct the initial ranking that is closer to the question than to the rationale). Then, a rationale is generated conditional con the reranked passages and input query.

**Strengths:**

Originality and significance: the authors' contribution is a nice application of both distillation and retrieval or knowledge augmentation for LMs.
Clarity: the paper is well written and it is clear and easy to read.

**Weaknesses:**

From the experiments section, it is not clear what type of retriever the baseline methods that include knowledge augmentation, is it a dense retriever or BM25? If it is the former, which specific encoder/decoder are used and in which task they are fine-tuned on?
Here, I am assuming that the retriever is with BM25 in the baselines:

Such comparisons to more than one neural retrieval augmented language models are important to paint a full picture of the contribution. One possibility is to use NQ, TriviaQA, other Q&A evaluation datasets or the KILT datasets to understand whether both the retrieved passages and the rationale augmentation is useful for smaller LMs for simpler Q&A tasks besides the reasoning Q&A used in the paper.

Alternatively, comparisons to other retrieval-augmented language models using the same benchmarks included in the paper can also help to quantitatively assess its performance grounding the claims that the cited paper mentions (i.e. that these models are not good for complex reasoning tasks). --> This experiment has been done during the rebuttal

**Questions:**

Please mention the encoder that is used to initialise the reranker in the main text, I have to tease it out of the appendix.

Would it make sense to do a comparisons to augmentation with rationales plus filtering: Zelikman et al. (2022) STaR: Bootstrapping Reasoning With Reasoning or do we expect to always useful valid rationales?

**Limitations:**

The authors include a comprehensive potential societal impact section in the appendix.

---

> ### Author Rebuttal · Authors · 2023-08-06
>
> We sincerely thank you for your constructive and helpful comments. We initially address all your concerns and questions below:
>
> ---
>
> > W1. It is not clear what type of retriever the baseline methods that include knowledge augmentation.
>
> Thank you for pointing it out and we will include more details about the retriever in Section 5.1. As you mentioned, we use BM25 as the retriever for the baseline methods, as described in Section B of the Supplementary File.
>
> ---
>
> > W2. Evaluation of proposed method on simple QA datasets (NQ, TriviaQA, KILT)
>
> Thank you for your suggestion. However, we would like to emphasize that the primary target of our work is on reasoning distillation; therefore, we hope you understand that improving on the simple QA datasets, which may not require the complex reasoning ability of LMs, is neither our focus nor the scope of this work.
>
> ---
>
> > W3. Comparisons to other retrieval-augmented language models using the same benchmarks included in the paper can also help to quantitatively assess its performance grounding the claims that the cited paper mentions (i.e., that these models are not good for complex reasoning tasks)
>
> Thank you for your suggestion.
> We respectfully emphasize that our main argument is that **knowledge augmentation is important when conducting reasoning distillation** for fine-tuning small LMs, especially for knowledge-intensive reasoning tasks, to supplement the limited capacity of small LM for memorizing knowledge.
> Therefore, a knowledge-augmented (retrieval-augmented) LM is not a direct competitor with our proposed method.
> Rather, it is one of the possible backbone networks that we can use for reasoning distillation with our method KARD.
>
> Nevertheless, following your suggestion, we compared against  Atlas [1], which is a state-of-the-art open-source knowledge-augmented LM, on the MedQA-USMLE dataset to measure the capability of knowledge-augmented LM on the tasks requiring complex reasoning ability.
>
> In the experiment on the MedQA dataset, the Atlas-base model with 220M parameters shows accuracy of **31.03**, which is comparable to the fine-tuned Flan-T5 with 250M parameters but **significantly worse** than our method KARD with accuracy of **38.15**. This result implies that **Atlas has limited reasoning ability**, especially in the domain requiring expert knowledge, which emphasizes the importance of reasoning distillation. Motivated by our findings, incorporating reasoning distillation on knowledge-augmented LM can be a meaningful research direction for future work.
>
> [1] Izacard et al., Atlas: Few-shot Learning with Retrieval Augmented Language Models, 2022
>
>
> ---
>
> > Q1. Please mention the encoder that is used to initialize the reranker in the main text.
>
> Thank you for the suggestion. As described in lines 109-114 of the Supplementary File,  the encoders for MedQA and StrategyQA are initialized with BioLinkBERT-base and LinkBERT-base, respectively.  We will specify details about the reranker in Section 5.1 of the main paper.
>
> ---
>
> > Q2. Would it make sense to do a comparison to augmentation with rationales + filtering as done in Zelikman et al. (2022)?
>
> Thank you for the suggestion. We would like to emphasize that “filtering” is orthogonal to knowledge augmentation. We can apply any filtering method to our KARD. Furthermore, we have already done it in our experiments by removing wrong rationales generated from large language models following Ho et al. [2] (See lines 82-89 of Appendix).
>
> [1] Zelikman et al., Star: Bootstrapping reasoning with reasoning, NeurIPS 2022.
>
> [2] Ho et al., Large language models are reasoning teachers, ACL 2023.

---

> > ### Comment · Reviewer_Z8sP · 2023-08-21
> > **Rebuttal acknowledgement.**
> >
> > Thank you to the authors for answering my questions and for conducting further experiments that highlight their contribution's main strengths. Specifically, I found very useful to see the ablation were they used the PubMed corpus instead of wikipedia (and KARD performs worse), the comparison against Atlas and the comparison of KARD with reasoning distillation + RAG. I encourage you to include these in the camera ready version of the paper. For these reasons, I don't have any other concerns and I am increasing my score.

---

> > > ### Author Response · Authors · 2023-08-21
> > > **Thank you**
> > >
> > > We are glad that our rebuttal addressed all of your concerns, and we are happy to see the reviewer raised the score.
> > > As you suggested, we will include discussions and experimental results against other retrieval-augmented methods in a future revision.
> > >
> > > Once again, we thank the reviewer, as your insights are invaluable in guiding our revisions.
> > > We are heartened by your positive reception of our work and will update it following your suggestions.

---

### Official Review · Reviewer_AUtH · 2023-07-07

**Soundness:** 3 good
**Presentation:** 4 excellent
**Contribution:** 3 good
**Rating:** 7
**Confidence:** 5

**Summary:**

The paper focuses on distilling the chain-of-thought reasoning capability from large LMs to small LMs in knowledge-intensive tasks. Since small LMs do not encode sufficient knowledge required for reasoning, the paper proposes to augment small models with a knowledge retriever that obtains relevant documents for a given task. Experiments show that the proposed method leads to more successful knowledge distillation, especially when the LM size is smaller.

**Strengths:**

1.	The proposed idea is well-motivated and sound.
2.	Sufficient experiments and detailed analysis are provided to demonstrate the effectiveness of the method.


**Weaknesses:**

One concern I have is the consequence of using multiple rationales to train the small LM, since this would misguide the model to learn that the answer prediction does not rely on the rationale. This may further lead to shortcut reasoning. I would suggest using simulation-based metrics to evaluate the faithfulness of the rationales to see if this is the case. Or you can randomly corrupt the generated rationales and see if the answer prediction is affected.

**Questions:**

How do you obtain the rationales from the large LM? The chain-of-thought prompting asks the LM to generate the rationale and then the answer. Do you simply provide the answer to the large LM and ask it to rationalize the answer?

**Limitations:**

Limitations are well discussed in the paper.

---

> ### Author Rebuttal · Authors · 2023-08-06
>
> We sincerely thank you for your constructive and helpful comments. We initially address all your concerns and questions below:
>
> ---
>
> > W1. One concern: The consequence of using multiple rationales to train the small LM, since this would misguide the model to learn that answer prediction does not rely on the rationale. This may further lead to shortcut reasoning.
>
> Thank you for your insightful comment. We have also noticed this problem to some extent, therefore, we first filter some erroneous rationales which lead to false prediction, following procedures described in lines 82-89 of the Appendix. In other words, if a small Flan T5 model makes a wrong prediction with the rationale generated by ChatGPT, such rationale is discarded. Furthermore, we have conducted analysis on the rationale diversity in Table 2 of the main paper. Corresponding results show that increasing the diversity of rationale (i.e., using multiple rationales) leads to better performance of small language models.
>
> ---
>
> > Q1. How do you obtain the rationales from the large LM?
>
> We use chain-of-thought prompting to generate rationales from the large LM. We explain the detailed procedure in lines 82-89 of the Appendix and provide the example prompt in Tables 4 and 5 of the Appendix.

---

### Official Review · Reviewer_7NNq · 2023-07-09

**Soundness:** 3 good
**Presentation:** 3 good
**Contribution:** 3 good
**Rating:** 6
**Confidence:** 3

**Summary:**

This paper proposes a retrieval-augmented knowledge distillation approach for QA tasks. This approach, KARD, extends reasoning distillation, which uses an LLM such as GPT-3.5 as a teacher model and distills a student model by learning from question and rationale pairs (generative loss). KARD has a retriever that obtains relevant documents based on the rationale. The retrieved documents are used for training a student model. In addition, a reranker is independently trained to select more relevant documents.

This approach is evaluated on two QA datasets: one from the biomedical domain, MedQA-USMLE, and StrategyQA, which covers more general and popular entities. EN Wikipedia is used as a knowledge base throughout experiments. The proposed approach, KARD (reasoning distillation+reranker), is compared with prompting approaches (e.g., few-shor, CoT) and other finetuning-based approaches (e.g., standard FT and knowledge-augmented FT). KARD outperforms all baselines by various margins depending on the datasets and model sizes. In addition, the authors provide analyses on several factors such as the number of rationales.


**Strengths:**

- This work proposes KARD, which integrates retrieval and reranking modules into the distillation framework.
- The experimental results support the effectiveness of this approach particularly on the QA task from the medical domain.
- Overall, this paper is well-written and easy to follow. The detailed analysis on model configurations is provided.


**Weaknesses:**

- Although this approach outperforms all the baselines, its gains become marginal with large model sizes, specifically in the cases of MedQA-USMLE and StrategyQA. On the other hand, KARD demonstrates a performance comparable to that of the oracle, ChatGPT, on StrategyQA, but lags significantly behind on MedQA-USMLE. The inconsistency in these results presents a challenge in interpretation.

**Questions:**

- Related to the first point about weaknesses, I believe including an analysis of the datasets (for instance, entity types and the diversity of entities) could be helpful. This is because the two datasets seem sufficiently different, and these differences might explain the results.

---

> ### Author Rebuttal · Authors · 2023-08-06
>
> We sincerely thank you for your constructive and helpful comments. We initially address all your concerns and questions below:
>
> ---
>
> > W1. Gains become marginal with large model sizes.
>
> As stated in the main paper lines 108-110, memorization of training data is essential for achieving good performance in language tasks [1] and memorization capacity is proven to be proportional to the model size [2]. Thus, larger models with better memorization may less rely on the retrieved knowledge for answering questions, and thus gains from retrieval become small as the model size increases.  We empirically observe this pattern in our experiment and have clearly discussed it in lines 261-266.
>
> Nevertheless, the largest XL model augmented with silver documents still outperforms the model without any knowledge augmentation in the MedQA-USMLE dataset. It implies that there is a potential of larger gains with better retrieval models, which opens up a promising direction for future work.
>
> [1] Brown et al., When is memorization of irrelevant training data necessary for high-accuracy learning?, STOC 2021.
>
> [2] Kim et al., Provable memorization capacity of transformers, ICLR 2023.
>
> ---
>
> > W2. KARD lags significantly behind on MedQA-USMLE compared to ChatGPT but not much on StrategyQA. The inconsistency in these results presents a challenge in interpretation.
>
> We think that such a discrepancy comes from different characteristics between MedQA and StrategyQA (See examples in Tables 11 and 12 in the Appendix). Specifically, the questions in MedQA are typically longer and contain more entities compared to the ones in StrategyQA, which requires a language model to leverage a significant amount of more knowledge to answer the questions in MedQA. Thus it is challenging for the smaller language model with the limited knowledge capacity to outperform larger language models like ChatGPT in MedQA, even when it is coupled with the knowledge retrieval. However, since the model augmented with silver knowledge achieves better performance than the one with knowledge retrieved by our reranker, we believe the gap between small and large language models can be further reduced with better retrievers.
>
> ---
>
> > Q1. Regarding W2, including an analysis of the datasets could be helpful.
>
> Thank you for your valuable suggestion. We include statistics of each dataset including the average number of entities and words in each question as follows:
>
> | |MedQA-USMLE | StrategyQA|
> |:---:|:---:|:---:|
> |Average number of entities | 35.72 | 1.19 |
> |Average number of words | 133.59 | 10.6 |
>
> As we can observe, questions in MedQA-USMLE generally have longer questions having more entities than questions in StrategyQA.

---

### Official Review · Reviewer_RfCw · 2023-07-26

**Soundness:** 2 fair
**Presentation:** 3 good
**Contribution:** 2 fair
**Rating:** 4
**Confidence:** 4

**Summary:**

In this paper, we propose the KARD model for small model Q&A through knowledge distillation + KB retrieval. The authors show experimentally that the model can outperform other models of 3B using only 250M parameters.

**Strengths:**

1. a model of LLM knowledge distillation + KB retrieval is proposed.
2. KARD outperforms other fine-tuning models.
3. a neural reranker based on similarity of rationales and passages is proposed.
4. the paper is easy to read.

**Weaknesses:**

1. a problem of the paper is the experimental design. In Table 1, the authors do not compare other knowledge-augmented LMs. this leads to experimental comparisons that are inadequate and unfair.
2. The finding of the paper-enhancing model effectiveness through KB retrieval-is not that surprising. This leads to a possible lack of innovation throughout the paper.
3. for MedQA-USMLE, using wikipedia as KB may not be as effective as using specialized medical KB.

**Questions:**

1. What if KARD switches to use a medical KB for MedQA-USMLE?
2. What is the time cost to train a reranker, as you need to compute the similarity between each pair of Wiki articles and training samples.

**Limitations:**

The authors have discussed potential technical limitations.

---

> ### Author Rebuttal · Authors · 2023-08-06
>
> We sincerely thank you for your constructive comments. We faithfully addressed all your concerns and questions below:
>
> ---
>
> > W1. Comparison against other knowledge-augmented LMs.
>
> Thank you for your suggestion. Please note that our main argument is that **knowledge augmentation is important when conducting reasoning distillation** when finetuning small LMs, especially for knowledge-intensive reasoning tasks, to supplement the limited capacity of small LM for memorizing knowledge. Therefore, **a knowledge-augmented LM is not a direct competitor with our proposed method**. Rather, it is one of the possible base model that can benefit from reasoning distillation with our KARD.
>
> Nevertheless, following your suggestion, we compared against Atlas [1], which is a state-of-the-art open-source knowledge-augmented LM, on the MedQA-USMLE dataset to measure the capability of knowledge-augmented LM on the tasks that require complex reasoning ability. In the experiment on the MedQA dataset, the Atlas-base model with 220M parameters shows accuracy of **31.03**, which is  **significantly worse** than our method **KARD** with accuracy of **38.15**. It implies that **Atlas has limited reasoning ability**, especially in the domain requiring expert knowledge, which emphasizes the importance of reasoning distillation. Motivated by our findings, incorporating reasoning distillation on knowledge-augmented LM can be a meaningful research direction for future work.
>
> [1] Izacard et al., Atlas: Few-shot Learning with Retrieval Augmented Language Models, 2022
>
> ---
>
> > W2. The finding of the paper, enhancing model effectiveness through KB retrieval, is not that surprising. This leads to a possible lack of innovation throughout the paper.
>
> We respectfully disagree with reviewer's point about the lack of innovation in our work because **this comment seems to be based on a critical misunderstanding of our work.** Our finding is **not limited** to enhancing model effectiveness through KB retrieval, and our method is not a straightforward combination of KB retrieval and reasoning distillation. Rather, our main contribution is in identifying the underrepresented issue that arises in the **reasoning distillation paradigm** due to the limited capacity of small language models to memorize knowledge, as clearly indicated in lines 42-43 of the main paper and identified by other reviewers (jTmB, 7NNq, AUtH).
>
> Also, our method is motivated by our theoretical analysis, which shows that knowledge augmentation with retrieval reduces the amount of memorization to perform well on knowledge-intensive reasoning tasks, and we find that the knowledge augmentation during the reasoning distillation enhances the performance of small models in knowledge-intensive reasoning tasks, especially in the domains that require knowledge not included in general LMs, such as the medical domain. Furthermore, we also introduce a novel reranker to retrieve documents relevant to generating rationales that lead to answering correct answers.
>
> ---
>
> > W3 & Q1. Using Wikipedia as KB may not be as effective as using specialized medical KB. What if KARD switches to use a medical KB for MedQA-USMLE?
>
> We use Wikipedia as a KB in experiments due to its generalizability across diverse domains. Furthermore, since Wikipedia also contains in-depth information about the medical and general healthcare domain, using Wikipedia as a KB for medical-oriented tasks is not very limiting (reference: https://en.wikipedia.org/wiki/Health_information_on_Wikipedia).  Please refer to the example in Appendix Table 11, where the passage includes compound medical knowledge though it is retrieved from Wikipedia.
>
> Moreover, following your suggestion, we experiment with PubMed as a knowledge base and summarize the results in the below table (initial experimental results already exist in Table 1 of Supplementary File). The results show that KARD with PubMed outperforms standard reasoning distillation but underperforms KARD with Wikipedia due to the limited ability of BM25 to retrieve relevant documents from the medical domain corpus.
>
> Specifically, considering the remarkable performance gain of  KARD with silver knowledge compared to BM25, the PubMed knowledge base is a valuable resource for knowledge augmentation. However, BM25 with a question as a query struggles to retrieve relevant documents from Pubmed's knowledge base. This observation shows a future research direction on the retrieval method tailored to the medical knowledge base.
>
> | Flan-T5 Base           | Wikipedia | PubMed |
> |------------------------|:---------:|:------:|
> | Reasoning Distillation |   31.03   |  31.03 |
> | KARD (BM25)            |   33.14   |  31.58 |
> | KARD (Reranker)        |   38.15   |  36.84 |
> | KARD (Silver knowledge) |   40.30   |  45.48 |
>
> ---
>
> > Q2. What is the time cost to train a reranker for computing the similarity between each pair of Wiki articles and training samples?
>
> In the training phase of the reranker, the initial step involves retrieving a set of candidate documents following the procedure outlined in lines 205-207 of the main paper, where the retriever takes about a second per a single training instance. Importantly, we can cache these candidate documents before the reranker training begins. As a result, there is no extra cost for computing similarity between all pairs of Wiki articles and training samples during reranker training.
>
> In other words, we only need to load cached candidate documents for each instance in the minibatch. We summarize the wall-time for each operation in a single training iteration below to show that the time consumption for loading cached candidate documents is negligible compared to forward and backward computation for model training.
>
> |                    | data load | forward | backward |
> |--------------------|-----------|---------|----------|
> | Wall time (second) | 0.01      | 0.9     | 0.14     |
> | Ratio              | ~1%       | 85.7%   | 13.3%    |

---

> > ### Comment · Reviewer_RfCw · 2023-08-18
> >
> > Thanks to the author for the reply. My main concern is about W2.
> >
> > Reasoning distillation is an important problem. Based on this, introducing the paradigm of knowledge augmentation is an innovation. However, I feel that this innovation is not hard to think of. I hope the author can answer the following questions:
> >
> > 1. Is introducing knowledge augmentation to reasoning distillation a challenging problem? What are the new challenges compared to traditional retrieval augmentation? Based on my understanding, the author's approach is similar to traditional retrieval augmentation methods, except that the knowledge is generated by a Large LM.
> >
> > 2. For the paradigm of knowledge augmentation in reasoning distillation, what new insights or inspirations can this paper offer to readers? As I said, introducing knowledge augmentation in reasoning distillation can enhance the performance, which is not something that is particularly surprising.

---

> > > ### Author Response · Authors · 2023-08-19
> > > **Response by Authors (1/2)**
> > >
> > > Thank you for reading our response and then providing additional comments on it. We sincerely appreciate your follow-up response and we would like to address your remaining concerns below.
> > >
> > > First of all, thank you for acknowledging that **reasoning distillation is an important problem**, and our contribution on introducing the paradigm of **knowledge augmentation into reasoning distillation is an innovation**. On the other hand, we understand your concern that this innovation may not be hard to think of, since there are some prior works on knowledge augmentation methods for language models. We answer your questions to address your concerns below.
> > >
> > > ---
> > >
> > > > **Q1.** Is introducing knowledge augmentation to reasoning distillation a challenging problem?
> > >
> > > Yes. Introducing knowledge augmentation to reasoning distillation is a challenging problem, which is significantly different from the challenge of previous retrieval-augmented methods.
> > >
> > > First of all, the core idea and challenge of reasoning distillation is to train the small language model to generate “rationales” generated by large language models.
> > > Regarding this challenge, we theoretically motivate that the existing naive reasoning distillation is suboptimal for knowledge-intensive tasks due to the lack of memorization capacity of small LMs. At this point, to supplement memorization capacity, our method needs an additional module to retrieve the **relevant knowledge from the external Knowledge Base (KB) to generate the high-quality rationale**. We believe this aspect significantly distinguish our method from existing retrieval augmented methods.
> > >
> > > In particular, as we discussed in lines 96-100, previous methods like RAG [1] do not consider intermediate rationales which are crucial for complex knowledge-intensive tasks. Rather, they retrieve documents with a question as a query, and a language model generates an answer based on the question and retrieved documents in both training and inference times. However, in our experiments, we empirically found that it is difficult to retrieve appropriate documents with a question as a query, which implies that existing knowledge augmentation methods (e.g., Atlas/RAG) can underperform our method in reasoning distillation (see below paragraphs for experimental comparison against RAG).
> > >
> > > In contrast, in our knowledge-augmented reasoning distillation, we can obtain the helpful knowledge to generate the rationale by retrieving the document with the **rationale** generated by the LLM as a query for the retriever during the training. However, as we discussed in lines 100-106 of the main paper, **this approach poses a new challenge** that the retriever should retrieve the helpful knowledge to generate the rationale with the “question” as a query in the inference stage, in order to generate meaningful rationales. To address this novel challenge, we introduce the algorithm to train the neural reranker tailored to knowledge-augmented reasoning distillation in Section 4.2 and empirically show clear effectiveness of  our proposed reranker.
> > >
> > > Furthermore, we agree that traditional retrieval-augmented methods are applicable to reasoning distillation and some readers might be curious about the effectiveness of traditional retrieval-augmented methods in reasoning distillation. To address this question, **we additionally conduct experiments on reasoning distillation with RAG** on two datasets we used in our previous experiments with the base-sized model having 250M parameters and then present their results in the below table.
> > >
> > > | (Flan-)T5 Base         | MedQA-USMLE | StrategyQA |
> > > |---|:---:|:---:|
> > > | Knowledge-augmented Fine-tuning |   33.39   |  52.11 |
> > > | RAG + Reasoning Distillation  |   24.84   |  54.24  |
> > > | KARD (Reranker)        |   **38.15**   |  **56.57**  |
> > >
> > > From the above empirical observation, we can emphasize that our method has significant advantages compared to the traditional retrieval augmented methods for reasoning distillation from the following three viewpoints:
> > >
> > > - Our knowledge augmentation method is **more effective** in reasoning distillation than the traditional methods, as KARD outperforms RAG in both datasets in performance.
> > > - Our method is **more efficient** than the traditional methods. We emphasize that RAG costs roughly **8 times more** computational budget (GPU memory) than KARD since it needs marginalization over retrieved documents in training.
> > > - Our method is **more versatile** than the traditional methods. As clearly shown in experiments on our answer to W1 of Reviewer jTmB, our method is **applicable to both encoder-decoder (T5) and decoder-only (GPT) language models** in contrast to RAG which is only applicable to encoder-decoder language models.
> > >
> > > We will include this discussion in the future revision.
> > >
> > > [1] Lewis et al., Retrieval-Augmented Generation for Knowledge-Intensive NLP Tasks, NeurIPS 2020.

---

> > > > ### Author Response · Authors · 2023-08-19
> > > > **Response by Authors (2/2)**
> > > >
> > > > > **Q2.** For the paradigm of knowledge augmentation in reasoning distillation, what new insights or inspirations can this paper offer to readers?
> > > >
> > > > In this work, we can offer new insights that, in reasoning distillation, supplementing the knowledge of small LMs with the knowledge retrieved from external knowledge bases is required to generate high-quality rationales for knowledge-intensive tasks based on both theoretical and empirical results (Section 3, 5).
> > > >
> > > > Furthermore, we offer new findings that, for a rational generation with knowledge augmentation, using the question as a query to retrieve the knowledge is less beneficial than the generated rationale. However, since the generated rationale from large models is not available during the inference time, we propose the neural reranker to address this challenge. We believe that performance improvement itself can bring significant inspiration to the importance of our knowledge augmentation method for reasoning distillation.
> > > >
> > > > ---
> > > >
> > > > We hope that our response sufficiently addresses your concerns. If you have any other concerns or comments, feel free to let us know so that we can address them! Thank you so much for your effort and time in reviewing our work.

---

> > > ### Author Response · Authors · 2023-08-21
> > > **Thank you for the score adjustment**
> > >
> > > We feel deeply grateful that the reviewer has read our response and adjusted the score accordingly.
> > >
> > > We would greatly appreciate any feedback on the remaining concerns, so that we can discuss and possibly further improve our work.
> > >
> > > Thank you again for your insightful questions and constructive suggestions. We are looking forward to your response.

---

### Official Review · Reviewer_jTmB · 2023-07-26

**Soundness:** 4 excellent
**Presentation:** 4 excellent
**Contribution:** 2 fair
**Rating:** 7
**Confidence:** 4

**Summary:**

The paper deals with the challenge of utilizing small LMs in knowledge intensive tasks. As recent LLMs have shown promising capabilities in tasks that require reasoning, however, deployment of such model can remain limited due to cost or data limitations. Thus, the authors turn to face the challenge of reasoning distillation to other LMs, mainly smaller LMs that might be more feasible for deployment. However, small LLMs are limited by their inferior limited capacity of knowledge and understanding, therefore an external source of relevant context could be utilized for bridging this gap.

The authors present Knowledge augmented reasoning distillation (KARD) that fine-tunes smaller LMs to generate rationals, with the aid of a LLM with high reasoning capabilities, and to augment its knowledge with relevant contexts from an external knowledge base, to answer complex questions. KARD utilized baseline approaches to tackle knowledge intensive tasks, such as CoT prompting to the LLM and fine-tuning a neural document re-ranker to retrieve high quality contexts during inference when access to a large LLM is unavailable.

The authors test their system on two knowledge intensive benchmarks that require some level of reasoning using one of multiple sources and/or steps (MedQA-USMLE and StrategyQA). The evaluations show that using KARD with a 250M parameter model shows superior performance compared to fine tuned models, knowledge augmented and in few-shot settings. Furthermore, the authors analyze how model size, training set size and reranker setup affects the performance of KARD.

The main contributions of the paper are: (1) novel method that combines reasoning distillation with knowledge augmentation using neural rerankers. (2) analysis showing that small LM (250M) are not sufficient for knowledge intensive tasks in domain specific cases. (3) KARD's performance vs. various baselines and technique is superior.

**Strengths:**

- The challenge presented is of high importance for real-world applications and has high value.
- Experimental setup, baselines chosen and evaluation results of KARD on MedQA-USMLE and StrategyQA are convincing and significant with clear accuracy improvement of smaller LMs (and also larger) compared to the different baselines. Proving the paper's main claim.
- Improving the neural re-ranker using the LLM's rationals aids KARD to find relevant contexts during inference, and beyond naive approaches like BM25.
- Well written with clear problem presentation, concept explanations, mathematical notations and definitions. It is easy to follow the concepts in the paper.

**Weaknesses:**

- Limited evaluation on different datasets (only 2) and models (T5/Flan-T5). In addition, performance of KARD methods vs. reasoning distillation in the StrategyQA task is limited compared to MedQA. The authors address this in the limitations section.
- A generic external knowledge base (wikipedia) that might not be suitable for medical-oriented tasks such as MedQA for example. We can further see evidence to this looking at the KARD (silver knowledge, oracle) evaluation where the gap between that model and with the best KARD is smaller vs. the large LLM (chatGPT).
- Somewhat limited development (or naive approach) of the main contribution of the paper, that is the combination of neural ranker with reasoning distillation. To the authors credit they have addressed this in the paper. The authors address this in the limitations section.
- Authors have addressed but did not evaluate a joint objective functions of neural ranker + distillation.
- Despite authors addressing many of the paper's limitations, I find it significantly hinders the reliability of work done in the paper.


**Questions:**

- The objective functions of the ranker and distillation (L_rerank and L_distill_kb) are independent. Why is that? have you evaluated combining both model updates?

**Limitations:**

The authors broadly addressed the major weaknesses and limitations of the paper.

---

> ### Author Rebuttal · Authors · 2023-08-06
>
> We appreciate your time and effort in providing constructive feedback, and we address your concern and questions below.
>
> ---
>
> > W1 & W6. Limited evaluation which significantly hinders the reliability of work
>
> We perform additional experiments on another dataset OpenbookQA [1] and with another decoder-only language model (OPT [2]). As shown in the below table, our KARD outperforms other baselines in those experimental settings, showcasing its generalizability.
>
> ||OpenbookQA | OpenbookQA | OpenbookQA | MedQA-USMLE |
> |:---|:---:|:---:|:---:|:---:|
> ||T5-base| T5-large | T5-xl | OPT-350M |
> |Fine-tuning|54.0|62.0|74.6|26.47|
> |Knowledge-augmented FT|53.8|64.6|73.8|25.84|
> |Reasoning Distillation|58.2|65.8|76.2|28.67|
> |KARD (BM25)|55.4|65.4|75.6|31.26|
> |KARD (Reranker)|**59.2**|**66.2**|**78.6**|**34.25**|
>
> [1] Mihaylov et al., Can a Suit of Armor Conduct Electricity? A New Dataset for Open Book Question Answering, EMNLP 2018.
>
> [2] Zhang et al., OPT: Open Pre-trained Transformer Language Models, preprint 2022
>
> ---
>
> > W2. The performance improvement is limited in the StrategyQA task compared to the MedQA
>
> We believe the performance improvement in StrategyQA task is significant, considering that KARD enables small models to achieve performance comparable to models with rationales from ChatGPT, especially in Large and XL size models.
> The performance gain may seem limited due to a smaller performance gap between the model without any knowledge augmentation and the Oracle model on StrategyQA compared to MedQA.
>
> ---
>
> > W3. A generic external knowledge base (wikipedia) may not be suitable for the medical-oriented tasks.
>
> We use Wikipedia as a KB in experiments due to its generalizability across diverse domains. Furthermore, since Wikipedia also contains in-depth information about the medical and general healthcare domain, using Wikipedia as a KB for medical-oriented tasks is suitable (reference: https://en.wikipedia.org/wiki/Health_information_on_Wikipedia).  Please refer to the example in Appendix Table 11, where the passage includes compound medical knowledge though it is retrieved from Wikipedia.
>
>
> Moreover, we experiment with PubMed as a knowledge base and show the results in the below table (initial experimental results already exist in Table 1 of Supplementary File). The results show that KARD with Pubmed outperforms standard reasoning distillation but underperforms KARD with Wikipedia due to the limited retrieval with BM25 on medical-domain passages.
>
> Specifically, considering the huge gap between KARD with BM25 and KARD with silver knowledge performance, PubMed’s knowledge base contains highly informative passages. However, BM25 with a question as a query has a limited ability to retrieve them, which results in worse candidate documents being reranked than the candidate documents retrieved from Wikipedia.  This observation suggests a future research direction on the retrieval method tailored to the medical knowledge base.
>
> | Flan-T5 Base           | Wikipedia | PubMed |
> |---|:---:|:---:|
> | Reasoning Distillation |   31.03   |  31.03 |
> | KARD (BM25)            |   33.14   |  31.58 |
> | KARD (Reranker)        |   38.15   |  36.84 |
> | KARD (Silver knowledge, oracle)|   40.30   |  45.48 |
>
> ---
>
> > W4. Somewhat limited development (or naive approach) of the main contribution of the paper.
>
> The main focus and contribution of our paper is **our novel knowledge-augmented reasoning distillation and reranking methods** with their supporting theoretical and empirical results.
>
> In particular, our main contribution is to **identify the issue arising in the reasoning distillation paradigm** due to the limited capacity of small language models to memorize knowledge. We propose an effective knowledge-augmented reasoning distillation to tackle the issue. Specifically, motivated by our theoretical analysis that knowledge augmentation with a retrieval reduces the amount of memorization to perform well on knowledge-intensive reasoning tasks, we retrieve a set of relevant documents that can guide the language model to correctly answer questions during the reasoning distillation. Lastly, we introduce a neural reranker to retrieve documents relevant to generating rationales that lead to answering correct answers.
>
> ---
>
> > W5 & Q1. Reasons for independent training of small LMs and reranker, not joint training.
>
> We would like to emphasize that we do not have to jointly train the reranker when distilling larger language models into smaller models. In particular, when distilling models with our method, we use the rationale that is generated from the large language model and the silver knowledge that is retrieved with the generated rationale (See Lines 187-190); therefore, the **reranking process is not involved during distillation**. On the other hand, our neural reranker is also trained separately by assigning high relevance scores to documents that are related to silver knowledge.
>
> Moreover, it is not straightforward to train the neural reranker along with the loss from distilling language models since reranking itself and choosing relevant documents are not differentiable operations. To enable it, we can alternatively utilize reinforcement learning algorithms [1], however, they are known to suffer from high variances. Also, if we consider retrieved documents as latent variables, we can approximate the marginal likelihood of training objectives with the retrieved top-k documents as done in the previous work [2]. However, since we need a large number of k different documents for better approximation of marginalization during training, it is computationally prohibitive to jointly train the retriever and the distilled language model. We leave addressing such technical challenges as future work.
>
> [1] Wang et al. R^3: Reinforced ranker-reader for open-domain question answering AAAI 2018.
>
> [2] Lewis et al., Retrieval-Augmented Generation for Knowledge-Intensive NLP Tasks, NeurIPS 2020.

---

### Author Rebuttal · Authors · 2023-08-08

Dear Reviewers,

Thank you for your considerable efforts in reviewing our paper and providing insightful reviews to our work. We appreciate that reviewers find our proposed idea well-motivated and sound.

We have responded to the individual comments from the reviewers below, and believe that we have successfully addressed all of them. To summarize our response,

- **[jTmB]** We include additional experimental results on one more dataset (OpenbookQA [1]) and a decoder-only pre-trained language model (OPT [2]).
- **[RfCw, Z8sP]** We discuss the relation of our work against a knowledge-augmented language model, including a comparison with Atlas [3].
- **[jTmB, RfCW]** We discuss the use of the medical knowledge base (PubMed corpus) as an alternative to Wikipedia for the MedQA-USMLE task.
- **[jTmB, 7NNq]** We include more discussions and analyses on the experimental results of Table 1.
- **[7NNq]** We conduct an analysis of the dataset statistics to compare the different features of both datasets used in experiments.
- **[RfCw]** We analyze and discuss the time cost to train a reranker.

We respectfully recommend the reviewer read our response and leave a comment if any concern or question still remains.

Sincerely,
Authors

[1] Mihaylov et al., Can a Suit of Armor Conduct Electricity? A New Dataset for Open Book Question Answering, EMNLP 2018.

[2] Zhang et al., OPT: Open Pre-trained Transformer Language Models, preprint 2022

[3] Izacard et al., Atlas: Few-shot Learning with Retrieval Augmented Language Models, 2022

---

### Decision · Program_Chairs · 2023-09-21

**Decision:**

Accept (poster)

**Comment:**

Minimizing model size with knowledge augmentation is a key challenge in deployment of deep models.  Previous works [26, 15] have shown that reasoning distillation is less effective for knowledge-intensive reasoning tasks [13] where factual knowledge is important. This work proposes a method that fine-tunes small LMs to generate rationales with augmented knowledge retrieved from an external knowledge base. More specifically it combines reasoning distillation [13] with knowledge augmentation [14] using neural rerankers.

Experiment shows that small T5 and Flan-T5 models on the knowledge-intensive reasoning datasets (MedQA-USMLE, StrategyQA) outperforms a 3B (12 times larger) model.

Reviewers are happy with the motivation of the approach  (combining LLM distillation and retrieval), the clarity of the writing, and the significance of this work. They also feel that some limitations still remains
1. The KB is generic (wikipedia) not a more realistic domain specific KB.
2. The baselines do not include many ablation results making it hard to judge the exact contribution and novelty of specific techniques in this work.